# ON THE REGULARIZATION LANDSCAPE FOR THE LINEAR RECOMMENDATION MODELS

## ABSTRACT

Recently, a wide range of recommendation algorithms inspired by deep learning techniques have emerged as the performance leaders on several standard recommendation benchmarks. While these algorithms were built on different deep learning techniques (e.g., dropouts, autoencoder), they have similar performance and even similar cost functions. This paper studies whether the models' comparable performance are sheer coincidence, or they can be unified under a single framework. We find that all competitive linear models effectively add only a nuclear-norm regularizer, or a Frobenius-norm regularizer. The former ones possess a (surprising) rigid structure that limits the models' predictive power but their solutions are low rank and have closed-form. The latter ones are more expressive so they have better performance. However, the performance leaders EASE and EDLAE have only full-rank closed-form solutions. Their low-rank counterparts can be solved only by hard-to-tune numeric procedures such as Alternating Directions Method of Multipliers (ADMM). Along this line of findings, we further propose two low-rank, closed-form solutions, derived from carefully generalizing Frobenius-norm regularization techniques.

## 1 INTRODUCTION

Research progress on algorithms for recommendation has escalated in recent years, partially fueled by the adoption of deep learning (DL) techniques. However, recent studies have found that many new deep learning recommendation models have shown sub-par performance against simpler linear recommendation models (Dacrema et al., 2019; Rendle et al., 2019). Although some studies are available to analyze linear vs non-linear models (Dacrema et al., 2019), it remains puzzling why these seemingly different techniques all result in models with similar performance or even similar cost functions. In the latest study, Jin et al. (2021) examine the relationship between the widely used matrix factorization (MF), such as ALS (Hu et al., 2008), and the linear autoencoder (LAE) which encompasses the recent performance leaders, such as EASE (Steck, 2019) and EDLAE (Steck, 2020). They considered two basic regularization forms (see Eqs. (1) and (6)) and found that the optimal (closed-form) solutions of both models recover the directions of principal components, while shrinking the corresponding singular values differently. They suggest that the difference may enable LAE to utilize a larger number of latent dimensions to improve recommendation accuracy. This finding highlights the similarity and difference between LAE and MF.

In this paper, we go much beyond the two basic models studied in (Jin et al., 2021), to analyze a large number of recent performance leaders of (linear) recommendation algorithms, and aim to provide an in-depth understanding of various regularized objective functions and towards unifying them through their closed-form solutions. In return, the closed-forms serve as the barebones engine to help reveal what drives the performance improvement for the recent recommendation algorithms.

We examine three open and closely related questions: **(i) Characterization of models:** Recent recommendation models are built upon a diverse set of techniques, including dropouts (Srivastava et al., 2014), variational methods (Kingma & Welling, 2014), and matrix denoising (Tipping & Bishop, 1999b). We aim to identify a unified regularization framework to (re)-interpret different models and understand how they are related. We are specifically interested in how models based on variational autoencoder (VAE, a natural generalization of recent approaches) relate to other models (Liang et al., 2018). **(ii) Weighted generalization of regularizers:** A key idea in a recent performance leader EDLAE (Steck, 2020) is to utilize of *weighted/non-uniform regularizers* on the

parameters' Frobenius norm. Applying dropouts(Cavazza et al., 2018) is shown to be equivalent to re-weighting the exponents in the regularizers, such as designing the weighted sum of regularizers based on other norms or tuning the exponent weights. We are also interested in determining what circumstance closed-form solutions still exist. **(iii) Low-rank closed-form solutions for EDLAE.** While most linear recommendation models are shown to have closed-form solutions, the low-rank version of recent performance leader EDLAE (Steck, 2020) remains an exception. Fitting this model requires using ADMM (Steck, 2020). Although its performance may not be better than its full-rank counterpart, low-rank solutions are easier to interpret, use less storage, and can be more scalable with respect to the number of items. In addition, a closed-form solution disentangles key performance drivers from nuances (e.g., need to tune learning rate or deal with local optimal), and can help reveal the key driven factors comparing with other closed-form solutions. More importantly, such solutions are significantly easier to implement and be tested using a generic matrix computation platform (without specialized recommendation library). Can we approximate low-rank EDLAE with closed-form solutions?

Our investigation led to the following discovery and resolves the above questions:

**Regularizer dichotomy (Section 2):** We found that all of the leading (linear) recommendation models can be categorized into those that implement possibly weighted *nuclear-norm* regularizers, or those that implement *Frobenius-norm* regularizers. Specifically, we characterize the Variational Linear Autoencoders (VLAE) as a form of the weighted nuclear-norm regularization problem, in which the weights possess a specific combinatorial structure. We observe that it is not matrix factorization or LAE that determines the shrinkage structure (as Jin et al. (2021) suggested), but instead it is the form of regularization. Thus, our paper provides a more complete and accurate characterization of a linear recommendation model's performance under different regularizations.

**Rigidity of nuclear-norm regularizers (Section 3):** With the dichotomy result, we next aim to understand whether the weighted sum idea for Frobenius-norm regularizers (Steck, 2020; Jin et al., 2021) can be generalized to nuclear-norm regularizers, and whether tuning exponent weights can be beneficial. First, while VLAE is equivalent to nuclear-norm regularization and is easier to optimize (weighted nuclear norms are non-smooth but VLAE's objectives are smooth), its closed-form solution possesses a (surprisingly) rigid structure, i.e., the weighted regularization will lead to an auto-sorting singular value reduction and the larger single values tend to receive smaller reduction (*non-ascending reduction*). Second, it has been shown that the solution structure for a model with a squared nuclear-norm regularizer $\|W\|_*^2$ (i.e., dropout's equivalence) is strikingly similar to that for using $\|W\|_*$-regularizer (Gu et al., 2014). We generalize the result to show that the solution structures for $\|W\|_*^p$ are highly similar for all $p \geq 1$ (Regularization invariant/rigidity with respect to $p$). But when $p = 1, 2$, the solution and hyperparameters possess favorable properties which make hyperparameter searches easier. This also partially explains why only $p = 1, 2$ have been extensively considered. These rigidity properties severely limit the search space and explains why models that use only nuclear-norm regularizers share the same performance ceiling even when hyperparameters are extensively searched.

**Closed-form solution for low-rank EDLAE (Section 4):** The (weighted) Frobenius-norm regularizers $\|\Lambda W\|_F^2$ ($\Lambda$ is the hyperparameter diagonal matrix) are implemented in EASE (Steck, 2019) and EDLAE (Steck, 2020). These models produce closed-form full-rank estimators; and if the zero-diagonal constraint on $W$ is enforced, their singular vectors will no longer coincide with those of the data, and will deliver (slightly) better performance. However, no closed-form solution for the low-rank estimator is known and existing approaches rely on ADMM or Alternating Least Square (Steck, 2020). In this paper, we propose two low-rank, closed-form estimators that deliver comparable results to the full-rank models (EASE and full-rank EDLAE) as well as the ADMM-based solutions (Steck, 2020), and thus resolve the aforementioned third open question.

## 2 BACKGROUND AND REGULARIZER DICHOTOMY

This section first explains the background, and then give an overview of the dichotomy results. Some key theorems are deferred to Section 3 and Appendices.

**Background.** Recommendation algorithms can be categorized into explicit ones that aim to predict unseen ratings between a user and an item and implicit ones that aim to predict actions, such as user click or add-cart (Steck, 2019; Dacrema et al., 2019; Zhang et al., 2019). We focus on the implicit

Table 1: Investigating the closed/analytic solutions of linear models. $dMat(\cdot)$ denotes a diagonal matrix, $diag(X)$ is the vector on the diagonal of $X$. $\Lambda$ is the (hyperparameter) diagonal matrix as a coefficient of the regularization term. $W^*$ (or $P^*$, $Q^*$, etc) is the optimal solution for corresponding case, except for cases 9-12, where $\widehat{W}$ is the low-rank closed-form solution.

| | Model | Regularization | Solution |
|---|---|---|---|
| Nuclear norm | 1. Regularized PCA (Zheng et al., 2018) | $\min_{P,Q} \|X - PQ\|_F^2 + \lambda \cdot (\|P\|_F^2 + \|Q\|_F^2)$ | $X \overset{\text{SVD}}{=} U\Sigma V^T \quad \Omega = \sqrt{(\sigma_i - \lambda)_+}$ 
 $P^* = U_k \quad Q^* = \Omega V_k^T$ |
| | 2. MF dropout (Cavazza et al., 2018) | $\min_{P,Q,d} \|X - PQ\|_F^2 + d\frac{1-p}{p} \cdot \sum_{k=1}^{d} \|P_k\|_2^2 \cdot \|Q_k^T\|_2^2$ 
 $\min_Y \|X - Y\|_F^2 + \frac{1-p}{p}\|Y\|_*^2$ | $X \overset{\text{SVD}}{=} U\Sigma V^T$ 
 $Y^* = P^* \cdot Q^*$ 
 $= U \cdot S_\mu(\Sigma) \cdot V^T$ |
| | 3. WLAE (Bao et al., 2020) | $\min_{W_1,W_2} \|X - XW_1W_2\|_F^2 + \|W_1\Lambda^{\frac{1}{2}}\|_F^2 + \|\Lambda^{\frac{1}{2}}W_2\|_F^2,$ | $W_1^* = V(I - \Lambda S^{-2})^{\frac{1}{2}} P^T$ 
 $W_2^* = P(I - \Lambda S^{-2})^{\frac{1}{2}} V^T$ |
| | 4. VLAE (this paper) | $\min_{P,Q} \|X - PQ\|_F^2 + \|\Lambda^{1/2}Q\|_F^2 + \|P\Lambda^{1/2}\|_F^2$ 
 $\min_{A,B} \|X - XAB\|_F^2 + \|\Lambda B\|_F^2 + \|XA\|_F^2$ 
 $\min_{rank(W)\leq k} \|X - W\|_F^2 + 2\|W\|_{w,*}$ | $X \overset{\text{SVD}}{=} U\Sigma V^T$ 
 $P^* = U_k \cdot diag(\sqrt{\sigma_1 - \lambda_{(k)}}, \ldots, \sqrt{\sigma_k - \lambda_{(1)}}) \cdot \Omega$ 
 $Q^* = \Omega^T \cdot diag(\sqrt{\sigma_1 - \lambda_{(k)}}, \ldots, \sqrt{\sigma_k - \lambda_{(1)}}) \cdot V_k^T$ 
 $A^* = X^\dagger P^* \Lambda^{\frac{1}{2}} \quad B^* = \Lambda^{-\frac{1}{2}} Q^*$ |
| Frobenius norm | 5. EASE (full rank) (Steck, 2019) | $\min_W \|X - XW\|_F^2 + \lambda \cdot \|W\|_F^2$ 
 $s.t. \quad diag(W) = 0$ | $C = (X^TX + \lambda I)^{-1}$ 
 $W^* = I - C \cdot dMat(diag(1 \oslash C))$ |
| | 6. DLAE(full rank) (Steck, 2020) | $\min_W \|X - XW\|_F^2 + \|\Lambda^{1/2} \cdot W\|_F^2$ 
 $\Lambda = \frac{p}{1-p} dMat(diag(X^TX))$ | $W^* = (X^TX + \Lambda)^{-1} X^TX$ |
| | 7. EDLAE(full rank) (Steck, 2020) | $\min_W \|X - XW\|_F^2 + \|\Lambda^{1/2} \cdot W\|_F^2$ 
 $\Lambda = \frac{p}{1-p} dMat(diag(X^TX))$ 
 $s.t. \quad diag(W) = 0$ | $C = (X^TX + \Lambda)^{-1}$ 
 $W^* = I - C \cdot dMat(diag(1 \oslash C))$ |
| | 8. LRR (Jin et al., 2021) | $\min_{rank(W)\leq k} \|X - XW\|_F^2 + \|\Gamma W\|_F^2$ | $\overline{Y}^* = \overline{X}W^* \overset{\text{SVD}}{=} U\Sigma V$ 
 $\widehat{W} = (X^TX + \Gamma^T\Gamma)^{-1}X^TX(V_kV_k^T)$ |
| | 9. EDLAE-ADMM (Steck, 2020) | $\min_{A,B} \|X - X(AB - dMat(diag(AB)))\|_F^2$ 
 $+ \|\Lambda^{1/2} \cdot (AB - dMat(diag(AB)))\|_F^2$ | ADMM update $A, B$ |
| | 10. LR-DLAE (this paper) | $\min_{rank(W)\leq k} \|X - XW\|_F^2 + \|\Lambda^{1/2} \cdot W\|_F^2$ 
 $\Lambda = \frac{p}{1-p} dMat(diag(X^TX))$ | $W^* = (X^TX + \Lambda)^{-1} X^TX$ 
 $\overline{Y}^* = \overline{X}W^* \overset{\text{SVD}}{=} U\Sigma V^T$ 
 $\widehat{W} = W^*(V_kV_k^T)$ |
| | 11. LR-EDLAE-1 (this paper) | $\min_{rank(W')\leq k} \|X - XW\|_F^2 + \|\Lambda^{1/2} \cdot W\|_F^2$ 
 $W = W' - dMat(diag(W'))$ 
 $\Lambda = \frac{p}{1-p} dMat(diag(X^TX))$ | $C = (X^TX + \Lambda)^{-1}$ 
 $W^* = I - C \cdot dMat(diag(1 \oslash C))$ 
 $\overline{Y}^* = \overline{X}W^* \overset{\text{SVD}}{=} U\Sigma V^T$ 
 $\widehat{W} = W^*(V_kV_k^T)$ |
| | 12. LR-EDLAE-2 (this paper) | $\min_{rank(W')\leq k} \|X - XW\|_F^2 + \|\Lambda^{1/2} \cdot W\|_F^2$ 
 $W = W' - dMat(diag(W'))$ 
 $\Lambda = \frac{p}{1-p} dMat(diag(X^TX))$ | $C = (X^TX + \Lambda)^{-1}$ 
 $W^* = I - C \cdot dMat(diag(1 \oslash C))$ 
 $W^* \overset{\text{SVD}}{=} U\Sigma V^T \quad \widehat{W} = U_k\Sigma_k V_k^T$ |

problem because it is more economically relevant. Here, let $n$ be the number of items and $m$ be the number of users. We are given a binary matrix $X \in \{0,1\}^{m \times n}$ that represents the interaction between users and items so far, i.e., $X_{i,j} = 1$ iff user $i$ has purchased or made a rating on item $j$. Our goal is to produce a real-valued matrix $\hat{X}$, which we evaluate against future interactions using information retrieval (Top-$k$) metrics such as Recall or nDCG.

The problem is closely related to matrix completion (MC) because $X_{i,j} = 0$ can be viewed as "missing a data point". But MC's evaluation metric is mean-squared error and is different from ours (Candès & Tao, 2010). The connection between two problems results in models with similar objectives (Zheng et al., 2018). A technique developed for one problem often finds its counterpart for the other. Nevertheless, because of the difference in evaluation, efficacy of an algorithm for one problem does not imply its performance guarantee for the other. Thus, the non-overlapping component between two problems remains substantial. We also remark that our structural results on weighted nuclear-norm is new and applicable to MC.

## 2.1 DICHOTOMY OF THE MODELS

This section explains that *(i)* All linear recommendation models can be categorizeed into those that use nuclear-norm regularizers and those that use Frobenius norm (see also Table 1), and *(ii)* The form of regularization, instead of whether the problem shall be cast as matrix factorization or LAE, determines the shrinkage structure (Proposition 1).

**Nuclear-norm regularizers.** Let $X \in \mathbf{R}^{m \times n}$ be a matrix of rank at most $k$ with $k$ leading singular values being $\sigma_1(X) \geq \sigma_2(X) \geq \cdots \geq \sigma_k(X)$. Let $\omega = (\omega_1, \ldots, \omega_k) \in (\mathbf{R}^+)^k$. The weighted nuclear norm of $X$ with respect to $\omega$ is defined as $\|X\|_{\omega,*} = \sum_{i=1}^{k} \omega_i \cdot \sigma_i(X)$. This is a natural generalization of the weighted nuclear norm for low-rank matrices (Gu et al., 2014). Despite its name, the weighted nuclear norm is neither convex nor differentiable unless $\omega_i$'s are sorted in *descending* order (Chen et al., 2013; Iglesias et al., 2020).

Nuclear-norm regularizers perform $\ell_1$-shrinkage over the estimator's singular values, which resembles performing $\ell_1$-shrinkage for coefficients in a linear model in LASSO (Tibshirani, 1996). Therefore, nuclear-norm regularizers also promote sparsity over the solution's singular values (i.e., the solution is usually low rank). Algorithms below effectively add only a nuclear-norm regularizer to MF.

*A1. Regularized PCA* (Udell et al., 2016; Zheng et al., 2018) aims to solve

$$\min_{P,Q} \|X - PQ\|_F^2 + \lambda\|P\|_F^2 + \lambda\|Q\|_F^2 \tag{1}$$

where $\lambda$ is the hyperparameter. It has been known that Eq. (1) is equivalent to $\min_{\hat{X}} \|X - \hat{X}\|_F^2 + 2\lambda\|\hat{X}\|_*$. To solve Eq. (1), one can use factored gradient descent (Bhojanapalli et al., 2016) or directly derive its closed-form solution (Kunin et al., 2019), which involves computation of SVD of $X$ (see case 1 in Table 1).

*A2. Matrix Factorization with dropouts.* This approach uses $PQ$ ($P \in \mathbf{R}^{m \times k}$, $Q \in \mathbf{R}^{k \times n}$) to approximate $X$. A standard dropout technique is utilized when we train $P$ and $Q$. Cavazza et al. (2018) show that optimization with dropout is equivalent to solving $\min_{\hat{X}} \|X - \hat{X}\|_F^2 + \lambda\|\hat{X}\|_*^2$, and the closed-form solution is obtained by shrinking all singular values of $X$ by the same magnitude of $\mu$, which depends on the data $X$ and the choice of hyperparameter $\lambda$. See details in Appendix E.3.

The next two approaches leverage techniques from (variational) autoencoder. We show that they also effectively add variants of nuclear-norm regularizers although this may not be clear at the first glance.

*A3. Weighted Linear Autoencoder (WLAE).* Consider the following (non-uniform) weighted $\ell_2$-regularization (Bao et al., 2020):

$$\|X - XW_1W_2\|_F^2 + \|W_1\Lambda^{\frac{1}{2}}\|_F^2 + \|\Lambda^{\frac{1}{2}}W_2\|_F^2, \tag{2}$$

where $\Lambda$ is a hyperparameter diagonal matrix and $W_1$, $W_2$ denote the encoder and the decoder network respectively. Bao et al. (2020) have shown a closed-form solution for Eq. (2) with a specific of diagonal matrix $\Lambda = dMat(\lambda_1, \lambda_2, \cdots, \lambda_k)$ when the weight is non-descending: $\lambda_1 \leq \lambda_2 \leq \cdots \leq \lambda_k$. It remains unclear if a closed-form solution exists for an arbitrary weight order.

*A4. Variational Linear Autoencoders (VLAE).* While not explicitly studied before, it is also natural to consider linear simplification of Variational Autoencoders, such as Multi-VAE (Liang et al., 2018), which has shown to exhibit strong performance for recommendation. To optimize VLAE, we need to find the maximum likelihood estimation (MLE) for the probabilistic model

$$p(\mathbf{x} \mid \mathbf{z}) = \mathcal{N}\left(W\mathbf{z} + \boldsymbol{\mu}, \sigma^2 I\right) \text{ and } p(\mathbf{z} \mid \mathbf{x}) = \mathcal{N}(V(\mathbf{x} - \boldsymbol{\mu}), D) \tag{3}$$

where $V$, $W$ denote the encoder and the decoder networks, $\sigma$ is a parameter, and $D$ is a diagonal covariance matrix (for the data $X \in \mathbb{R}^{m \times n}$). We have the following observation.

**Lemma 1.** *Consider optimizing the ELBO (Evidence Lower Bound) (Kingma & Welling, 2014) for the above LVAE model (Eq. (3)). When $\boldsymbol{\mu} = 0$, this optimization problem is equivalent to minimizing*

$$\mathcal{L} = \|X - XV^{\mathrm{T}}W^{\mathrm{T}}\|_F^2 + m\|\sqrt{D}W^T\|_F^2 + \sigma^2\|XV^{\mathrm{T}}\|_F^2 + g(D, \sigma) \tag{4}$$

*where $g(D, \sigma) = -\sigma^2 m\left(\log |D| - \mathrm{tr}(D) + k - n\log(2\pi\sigma^2)\right)$.*

Here, we set $\boldsymbol{\mu} = 0$ for simplicity and following standard practices in recommendations. The proof can be found in Appendix D.1. We may further "clear up" Eq. (4) and obtain the following optimization problem (see Appendix D.3):

$$\min_{A \in \mathbf{R}^{n \times k}, B \in \mathbf{R}^{k \times n}} \|X - XAB\|_F^2 + \|XA\|_F^2 + \|\Lambda B\|_F^2, \tag{5}$$

in which decision variables are $A$ and $B$, and the hyperparameter is a diagonal matrix $\Lambda \in \mathbf{R}^{k \times k}$.

In Section 3, we show that solving Eq. (5) (A4) is equivalent to solving $\|X - W\|_F^2 + \lambda \|W\|_{\omega,*}$ subject to $\text{rank}(W) \leq k$, where $\omega$ consists of $\Lambda$'s diagonal values, *sorted in non-descending order.* Furthermore, we characterize the optimal solution for $\|X - W\|_F^2 + \lambda \|W\|_*^p$ for any $p \geq 1$. We shall show that regardless the choice of $p$, the optimal solution uniformly shrinks all $X$'s singular values by a constant magnitude $\mu$ (and to 0 if a singular value is already less than $\mu$). Finally, *for all nuclear-norm-based approaches discussed above, the estimators always keep singular vectors of $X$ and shrink its singular values.* Therefore, the solution space offered by nuclear-norm regularizers is quite constrained, which limits their predictive power.

**Frobenius-norm regularizers.** Most algorithms below were originally motivated by the design of (denoising) autoencoders. It has been shown that they effectively add a Frobenius-norm regularizer.

*A5. EASE (Steck, 2019)* aims to optimize $\min_W \|X - XW\|_F^2 + \lambda \cdot \|W\|_F^2$ subject to the constraint that $diag(W) = 0$. A closed-form solution exists for this problem (case 5 in Table 1).

*A6. DLAE (Steck, 2020)* adds a weighted Frobenius-norm regularizer so the objective becomes $\min_W \|X - XW\|_F^2 + \|\Lambda^{1/2} \cdot W\|_F^2$, where $\Lambda = \frac{p}{1-p} dMat(diag(X^T X))$, $dMat(\cdot)$ denotes a diagonal matrix, $diag(X)$ is the vector on the diagonal of $X$. The closed-form solution also exists (case 6 in Table 1).

*A7. EDLAE (Steck, 2020)* integrates weighted Frobenius norm in DLAE with EASE's diagonal constraint so its objective is the same as DLAE but it requires $diag(W) = 0$. A closed-form solution exists for this problem (case 7 in Table 1). For the low-rank EDLAE (case 9 in Table 1), an ADMM algorithm is used.

*A8. Tikhonov regularization/Low Rank Regression (LRR) (Jin et al., 2021).* Let $V_k$ be the $k$ leading right singular vectors of $X$. Jin et al. (2021) find an estimator that solves

$$W = \arg \min_{rank(W) \leq k} \|X - XW\|_F^2 + \|\Gamma W\|_F^2, \tag{6}$$

where $\Gamma = \Lambda^{\frac{1}{2}} V_k^T$ and $\Lambda = diag(\lambda'_1, \cdots, \lambda'_k)$ is a hyperparameter. The closed-form solution is

$$W^* = V_k \cdot dMat(\frac{\sigma_1^2}{\sigma_1^2 + \lambda'_1}, \cdots, \frac{\sigma_k^2}{\sigma_k^2 + \lambda'_k}) V_k^T \tag{7}$$

We first remark that A5 and A6 produce full-rank estimators. A7 can produce either full-rank or low-rank estimator (via ADMM) and has the best performance (among all approaches we discussed). The estimator from A8 is low-rank and has a closed-form solution but it has to keep singular vectors of $X$ so its predictive power is also limited. Nevertheless, A8 is conceptually interesting because it uses Frobenius-norm regularizers but its solution space cover the solution space offered in A4 (and thus also A1-A4).

**Proposition 1.** *For any regularized instances in the form of Eq. (5) with regularization parameter $\Lambda$ such that $\sigma_i(X) \geq \lambda_{(k-i)}$ for all $i$, there is a corresponding Tikhonov regularized instance with $\boldsymbol{\Gamma} = \Lambda^{\frac{1}{2}} V_k^{\mathrm{T}}$ that provides the same regularization effect.*

The proof is in Appendix D.2. A major implication of Proposition 1 is that *we can focus on designing Frobenius-norm regularizers because it also gets the value from using nuclear-norm regularizers.*

As mentioned earlier, no closed-form solution exists for the low-rank EDLAE (case 9 in Table 1) so existing approaches rely on ADMM or Alternating Least Square algorithms (Steck, 2020). In Section 4, we will introduce two low-rank models based on Frobenius-norm regularizers that have both competitive performance and closed-form solutions.

## 3   RIGIDITY OF NUCLEAR-NORM REGULARIZERS

This section presents two results. *(i)* Eq. (5) (A4) is equivalent to a model with weighted nuclear-norm, where the weights are diagonal values of $\Lambda$ arranged in non-descending order. Because weighted nuclear norms are non-differentiable in general so Eq. (5) compiles non-differentiable objectives into differentiable ones, which are easier to optimize. Also, the auto-sorting property restricts Eq. (5) from expressing an arbitrary weight sequence in $\|W\|_{\omega,*}$, which limits its predictive power. *(ii)* We give a closed-form solution for models with regularizer $\|W\|_*^p$ for all $p \geq 1$. The solutions share the same structure so tuning $p$ will not improve a model's predictive power.

**Rigidity of closed-form solutions.**   We first analyze Eq. (5) (A4). We start with the problem:

$$\min_{P,Q} \|X-PQ\|_F^2 + \|\Lambda^{\frac{1}{2}}Q\|_F^2 + \|P\,\Lambda^{\frac{1}{2}}\|_F^2, \text{ or equivalently, } \min_{P,Q} \|X-PQ\|_F^2 + \|\Lambda Q\|_F^2 + \|P\|_F^2, \quad (8)$$

Note that when we let $P' = XA^*$ and $Q' = B^*$, where $A^*$ and $B^*$ are an optimal solution of Eq. (5), the syntax of Eq. (8) matches with that of Eq. (5). This implies that the solution for Eq. (8) is a lower bound of that for Eq. (5). These two solutions coincide only when the columns in the optimal $P^*$ in Eq. (8) are spanned by the columns of $X$.

We next find a closed-form solution $(P^*, Q^*)$ for Eq. (8), and show that indeed the column space of $P^*$ is in the column space of $X$.

**Proposition 2.** *Let $f : \mathbf{R}^{m \times n} \to \mathbf{R}^+$ be any cost function. Let $P \in \mathbf{R}^{m \times k}$ and $Q \in \mathbf{R}^{k \times n}$. Let $\Lambda \in \mathbf{R}^{k \times k}$ be a diagonal matrix such that $\lambda_i = \Lambda_{ii} \geq 0$ ($i \in [k]$). Let also $\omega = (\lambda_{\pi(1)}, \lambda_{\pi(2)}, \ldots, \lambda_{\pi(k)})$, where $\pi$ is a permutation on $[k]$ such that $\lambda_{\pi(1)} \leq \lambda_{\pi(2)} \leq \cdots \leq \lambda_{\pi(k)}$. The following two optimization problems have the same optimal values*

$$OPT1: \quad \min_{P,Q} \quad f(PQ) + \|\Lambda^{\frac{1}{2}}Q\|_F^2 + \|P\Lambda^{\frac{1}{2}}\|_F^2.$$

$$OPT2: \quad \min_W \quad f(W) + 2\|W\|_{\omega,*}$$
$$\quad \text{subject to} \quad \text{rank}(W) \leq k.$$

*In addition, if $(P^*, Q^*)$ is an optimal solution for $OPT1$, then $W^* = P^*Q^*$ is an optimal solution for $OPT2$. If $W^*$ is an optimal solution for $OPT2$, then there exists an optimal solution $(P^*, Q^*)$ for $OPT1$ such that $W^* = P^*Q^*$.*

See Appendix D.4 and Appendix D.5 for the full analysis. We note that diagonals of $\Lambda$ do not have to be sorted in ascending order as stated in (Bao et al., 2020) because any permutation of the diagonals will be equivalent to $OPT2$. In addition, Proposition 2 is applicable to A3 & A4. $f(\cdot)$ in A3 & A4 is the reconstruction error, in which case closed-form solutions exist:

**Corollary 1.** *Let $\mathbf{X} \in \mathbf{R}^{m \times n}$ ($m \geq n$) be a full rank matrix. Let $\Lambda$ be a diagonal matrix with $\Lambda_{ii} \geq 0$ for all $i$. Let $P \in \mathbf{R}^{m \times k}$ and $Q \in \mathbf{R}^{k \times n}$. Consider the optimization problems*

$$\min_{P,Q} \|X - PQ\|_F^2 + \|\Lambda^{\frac{1}{2}}Q\|_F^2 + \|P\Lambda^{\frac{1}{2}}\|_F^2 \quad (9)$$

$$\min_{A,B} \|X - XAB\|_F^2 + \|\Lambda B\|_F^2 + \|XA\|_F^2. \quad (10)$$

*Let the SVD of $X$ be $U\Sigma V^{\mathrm{T}}$, and let $\Sigma_k \in \mathbf{R}^{k \times k}$ be a matrix comprising the $k$ largest singular values of $X$, and $U_k$ and $V_k$ be the corresponding singular vectors. Let $\lambda_{(1)} \geq \lambda_{(2)} \geq \cdots \geq \lambda_{(k)}$ be the sorted sequence of the diagonal values from $\Lambda$. Eq. (9) has a closed-form solution:*

$$P^* = U_k diag(\sqrt{(\sigma_1 - \lambda_{(k)})^+}), \ldots, \sqrt{(\sigma_k - \lambda_{(1)})^+})\Omega,$$
$$Q^* = \Omega^{\mathrm{T}} diag(\sqrt{(\sigma_1 - \lambda_{(k)})^+}), \ldots, \sqrt{(\sigma_k - \lambda_{(1)})^+})V_k^{\mathrm{T}}, \quad (11)$$

*where $\Omega$ is a unitary matrix that corresponds to the permutation $\pi$ such that $\lambda_{\pi(1)} \leq \cdots \leq \lambda_{\pi(k)}$. In addition, Eq. (10) has a closed-form solution: $A^* = X^{\dagger}P^*\Lambda^{\frac{1}{2}}$ and $B^* = \Lambda^{-\frac{1}{2}}Q^*$, where $X^{\dagger}$ is the pseudo-inverse of $X$.*

**Rigidity of $\|W\|_*^p$**   In (Cavazza et al., 2018), it was observed that when we use a neural net $X = PQ$ (with learnable parameters being $P$ and $Q$) to train a model and a standard dropout is used, the

objective is equivalent to solving $\min_W \|X - W\|_F^2 + \lambda \|W\|_*^2$. While the regularizer $\|W\|_*^2$ deviates from the standard one $\|W\|_*$, the optimal solution here is $W = US_\mu(\Sigma)V^T$, where $U\Sigma V^T$ is SVD of $X$, and $S_\mu(\Sigma)$ is a diagonal matrix such that its $(i,i)$-th element is $(\Sigma_{i,i} - \mu)^+$, in which $\mu$ depends on the data $X$ and $\lambda$. In other words, the optimal solutions for regularizers $\|W\|_*^2$ and $\|W\|_*$ are strikingly similar. Thus, we are interested in how regularizers with different exponents are connected. Our main observation is that for *any* regularizer $\|W\|_*^p$ $(p \geq 1)$, the optimal solution has the same structure.

**Lemma 2.** *Let $X \in \mathbf{R}^{m \times n}$, where $m \geq n$. Let the $d$ leading SVDs of $X$ be $U_d$ and $V_d$ respectively. Let $\sigma_1, \ldots, \sigma_n$ be the singular values of $X$. Consider the optimization problem:*

$$\min_W \frac{1}{2}\|X - W\|_F^2 + \lambda \|W\|_*^p. \tag{12}$$

*Let $\mu_k = \frac{1}{k}\sum_{i \leq k} \sigma_i(X)$, $\eta(\mu_k)$ be the positive root of the function $z + \lambda k z^{p-1} - k\mu_k$ and $d$ be the largest value such that $\sigma_d(X) - \lambda(\eta(\mu_d))^{p-1} \geq 0$. Let $\mu = \lambda(\eta(\mu_d))^{p-1}$. The optimal solution of $W$ is $U_d \cdot dMat(\sigma_1 - \mu, \sigma_2 - \mu, \ldots, \sigma_d - \mu) \cdot V_d^T$.*

Lemma 2 implies that regularizers $\|W\|_*^p$ with different $p$'s differ in how the shrinkage variable $\mu$ is obtained. Observing that $\mu$ is a hyperparameter that requires extensive tuning, all regularizers $\|W\|_*^p$ provide the same learning power. As noted earlier, $\mu$ is a function of the data $X$ unless $p = 1$; $\lambda$ needs to be rescaled when $X$ is scaled by a constant factor unless $p = 2$. This implies it could be easier to tune $\mu$ when $p = 1, 2$, and explains why only $p = 1, 2$ have been extensively considered.

## 4 LOW-RANK FROBENIUS-NORM REGULARIZATIONS

This section presents two closed-form low-rank estimators with competitive performance.

**Approximate low-rank DLAE and EDLAE.** Recall that (full-rank) DLAE solves:

$$\min_W \|X - XW\|_F^2 + \|\Lambda^{\frac{1}{2}}W\|_F^2 \quad s.t. \quad \Lambda = \frac{p}{1-p}dMat(diag(X^TX)) \tag{13}$$

and EDLAE with the additional $diag(W) = 0$ constraint. While both the regularizations use nuclear-norm and the full-rank DLAE/EDLAE solutions all have closed-form solutions, such solution is unknown for the low-rank DLAE and EDLAE, which still require ADMM (Steck, 2020). The closed-form solutions will help both better understand and compare these models, and determine the hyperparameters such as learning rates, which is usually difficult for the ADMM type solutions.

Low-rank DLAE is a special case of Tikhonov regularization (Eq. (6)) so it has a closed-form solution, namely LR-DLAE (Jin et al., 2021). See case 10 in Table 1. However, for EDLAE, it enforces the zero-diagonal constraint, which makes the exact solution difficult to express. Recall that that low-rank EDLAE aims to solve :

$$W = \arg\min_{rank(W) \leq k} \|X - X(W - dMat(diag(W)))\|_F^2 + \|\Lambda^{\frac{1}{2}}(W - dMat(diag(W)))\|_F^2 \tag{14}$$

Here, we present an approximate low-rank closed-form solution of Eq. (14) by decomposing the optimization problem into two subproblems, which is similar to (Jin et al., 2021). For detailed analysis and process, please refer to Appendix B. We first consider the full-rank closed-form solution for EDLAE (Eq. (13) with zero-diagonal constraint), which is given by:

$$W^* = I - C \cdot dMat(1 \oslash diag(C)), \text{ where } C = (X^TX + \Lambda)^{-1}$$

Then, we consider two (closed-form) approaches to produce low-rank matrix approximation of $W^*$:

**Method 1 (LR-EDLAE-1): Selecting $\widehat{W}$ to best approximate the performance of $W^*$ without the zero-diagonal constraint:**

$$\widehat{W} = \arg\min_{rank(W) \leq k} \|\overline{X}W^* - \overline{X}W\|_F^2$$
$$= \arg\min_{rank(W) \leq k} \|XW^* - XW\|_F^2 + \|\Lambda^{\frac{1}{2}}(W^* - W)\|_F^2 \tag{15}$$

where $\overline{X} = \begin{bmatrix} X \\ \Lambda^{\frac{1}{2}} \end{bmatrix}$. Noting, in the full-rank problem Eq. (13), it forces the diagonal of derived matrix $(W^*)$ to be zero. Here, we relax the zero-diagonal constraint. This relaxation corresponds to lower

bounding an approximate low-rank EDLAE objective function when rank $k$ is sufficiently large. (Details in Appendix B). The closed-from solution of Eq. (15) is given by:

$$\boxed{\widehat{W} = W^*(Q_k Q_k^T)} \tag{16}$$

where $Q_k$ takes the first $k$ rows of matrix $Q$: $\overline{X}W^* \stackrel{\text{SVD}}{=} P\Sigma Q^T$.

**Method 2 (LR-EDLAE-2): SVD approximation of $W^*$:** The alternative solution is to simply perform SVD: $W^* \stackrel{\text{SVD}}{=} U\Sigma V^T$, and which gives a low-rank estimation of $W^*$: $\widehat{W} = U_k \Sigma_k V_k^T$. This solution corresponds to upper bounding the minimization objective in Eq. (15) (Details in Appendix B).

Both approaches relax the hard zoer-diagonal constraint on $\widehat{W}$. In the next section, the experimental results show both approaches can provide comparable or better performance compared with the ADMM solution, and also very close to the full rank EDLAE solution.

## 5 EXPERIMENTAL RESULTS

This section experiments different regularizations for linear recommendation models, with a goal to validate the efficacy of various regularizations (all can be categorized under nuclear or Frobenius norms) together with their closed-form solutions. We answer three questions: **Q1.** How do the closed-form solutions of low-rank Frobenius-norm perform compared with the ADMM solutions (Section 4) and how does the weighted nuclear-norm regularizer for matrix factorization (Proposition 2 and Corollary 1) perform? **Q2.** What is the tradeoff between the number of factors (rank $k$) and the recommendation accuracy? **Q3.** How does the ordering of weights from small to large (non-descending) for adjusting the singular values (Corollary 1) affect the recommendation performance?

**Experimental Setup:** We use three commonly used datasets for recommendation studies: MovieLens 20 Mil. (ML-20M) (Harper & Konstan, 2015), Netflix Prize (Netflix) (Bennett et al., 2007), and the Million Song Data (MSD)(Bertin-Mahieux et al., 2011). We obtained the datasets and all benchmarks from authors of EASE (Steck, 2019), EDLAE (Steck, 2020), and Mult-VAE (Liang et al., 2018).

Similar to the latest study in EASE (Steck, 2019), and EDLAE (Steck, 2020), we consider the following state-of-the-art recommendation models: ALS (WMF) (Hu et al., 2008) for matrix factorization approaches, SLIM (Ning & Karypis, 2011), EASE (Steck, 2019), and EDLAE (Steck, 2020) for linear autoendoers, CDAE (Wu et al., 2016), Mult-DAE and Mult-VAE (Liang et al., 2018) for deep learning models. The experiment settings for these baseline are the same as (Liang et al., 2018; Steck, 2019; 2020). Also we follow their practice (Liang et al., 2018; Steck, 2019; 2020) for the *strong generalization* by splitting the users into training, validation and tests group, and report performance metrics $Recall@20$, $Recall@50$ and $nDCG@100$. Finally, we note that our code are openly available (see Appendix E.1).

**Q1: Low-Rank Frobenius-Norm and (Weighted) Nuclear-Norm Regularization:** We evaluate the low-rank Frobenius-norm-based regularization and the nuclear-norm-based regularization (Eq. (1)) for the matrix factorization. Here EDLAE-ADMM, LRR, LR-DLAE, LR-EDLAE-1, LR-EDLAE-2, MF dropout and VLAE are listed in Table 1. To determine the non-descending order of weights $\lambda_i$ for the closed-form solution in Eq. (11), we follow the practice in weighted nuclear-norm regularization in (Gu et al., 2014) as well as the optimized pPCA weight (Lucas et al., 2019b). Let $\lambda_i = \frac{C}{\sigma_i}$ where $C$ is a hyperparameter, and we perform grid-search to find the optimal one.

Table 2 shows that the weighted nuclear-norm regularization (VLAE) based matrix factorization performs worse than the constant weighted version (Regularized PCA). The latter demonstrates strong performance comparing against the WFM/ALS (one of the most popular implicit matrix factorization algorithms). The closed-form solutions (LR-DLAE, LR-EDLAE-1 and LR-EDLAE-2) all perform comparable with the ADMM-based low-rank solution and the full-rank DLAE and EDLAE solutions.

**Q2: nDCG vs Rank $k$ for low-rank Frobenius norm:** In this experiment, we focus on evaluating the recommendation accuracy (using nDCG) against the rank $k$. Specifically, we vary $k$ from around $1K$ to around $10K$, and we tune and compare four different methods, including EDLAE-ADMM, LR-DLAE, LR-EDLAE-1, and LR-EDLAE-2. We have the following observations from Fig. 1: 1) As $k$ increases, the recommendation accuracy also increases in general; however, most of them reach a plateau around similar $K$, and for different datasets, the saturating point varies. 2) LR-DLAE performs

Table 2: The performance comparison between different regularizations. We highlight the best results in bold and underline the 2nd best results for each metric. See Table 1 for the notation.

| Model | | | ML-20M | | | Netflix | | | MSD | | |
|---|---|---|---|---|---|---|---|---|---|---|---|
| | | | Recall@20 | Recall@50 | nDCG@100 | Recall@20 | Recall@50 | nDCG@100 | Recall@20 | Recall@50 | nDCG@100 |
| Frobenius Norm | full rank | EASE | 0.391 | 0.521 | 0.420 | 0.362 | 0.445 | 0.393 | 0.333 | 0.428 | 0.389 |
| | | DLAE | 0.392 | 0.527 | 0.424 | 0.362 | 0.446 | 0.395 | 0.329 | 0.426 | 0.387 |
| | | EDLAE | 0.393 | 0.523 | 0.424 | **0.366** | **0.449** | **0.398** | **0.334** | **0.429** | **0.392** |
| | low rank | EDLAE-ADMM | 0.392 | 0.524 | 0.424 | 0.365 | 0.448 | 0.396 | 0.330 | 0.424 | 0.386 |
| | | LRR | 0.376 | 0.511 | 0.408 | 0.348 | 0.431 | 0.380 | 0.248 | 0.335 | 0.301 |
| | | LR DLAE | 0.392 | 0.527 | 0.424 | 0.362 | 0.445 | 0.395 | 0.306 | 0.403 | 0.363 |
| | | LR-EDLAE-1 | 0.392 | 0.523 | 0.424 | 0.365 | **0.449** | **0.398** | 0.327 | 0.423 | 0.384 |
| | | LR-EDLAE-2 | 0.392 | 0.523 | 0.424 | 0.365 | **0.449** | **0.398** | 0.325 | 0.421 | 0.382 |
| Nuclear Norm | | MF dropout | 0.367 | 0.501 | 0.393 | 0.334 | 0.418 | 0.365 | 0.270 | 0.367 | 0.328 |
| | | Regularized PCA | 0.364 | 0.501 | 0.392 | 0.331 | 0.417 | 0.365 | 0.229 | 0.313 | 0.279 |
| | | VLAE | 0.348 | 0.474 | 0.378 | 0.325 | 0.405 | 0.357 | 0.205 | 0.254 | 0.286 |
| Baseline | | WMF/ALS | 0.360 | 0.498 | 0.386 | 0.316 | 0.404 | 0.351 | 0.211 | 0.312 | 0.257 |
| | | SLIM | 0.370 | 0.495 | 0.401 | 0.347 | 0.428 | 0.379 | no results in (Ning & Karypis, 2011) | | |
| | | CDAE | 0.391 | 0.523 | 0.418 | 0.343 | 0.428 | 0.376 | 0.188 | 0.283 | 0.237 |
| | | MULT-DAE | 0.387 | 0.524 | 0.419 | 0.344 | 0.438 | 0.380 | 0.266 | 0.363 | 0.313 |
| | | MULT-VAE | **0.395** | **0.537** | **0.426** | 0.351 | 0.444 | 0.386 | 0.266 | 0.364 | 0.316 |
| # items | | | 20108 | | | 17769 | | | 41140 | | |
| # users | | | 136677 | | | 463435 | | | 571353 | | |
| # interactions | | | 10million | | | 57million | | | 34million | | |

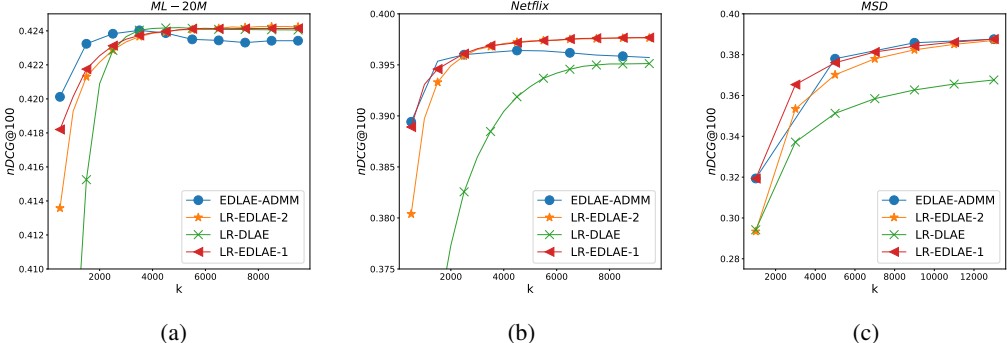

|     (a)     |     (b)     |     (c)     |

Figure 1: Low-rank models $nDCG@100$ on test data for 3 datasets.

worse than EDLAE-based approaches in two out of three datasets. This partially demonstrates the benefits of zero-diagonal constraint. 3) The closed-form solution of EDLAE performs comparable or even slightly better than ADMM methods as $K$ grows; but when $K$ is relatively small, ADMM method performs slightly better. But none-the-less, for most of the reasonable choices of $k$ when low rank approximates full rank, the closed-form solution performs comparable or better.

**Q3: Impact of Weight Ordering:** Finally, we study how the ordering of weights from small to large (non-descending) for adjusting the singular values ( Corollary 1) affects the recommendation performance using matrix factorization (closed-form solution in Eq. (11)). Our results are in Table 3. Here, we obtain the searched optimal weight parameters from weighted Tikhonov regularization (following the approach in (Jin et al., 2021)), and map it back to the parameters in the closed-form solution ( Proposition 1). Then we sort the parameters in the non-descending order, and then report their results in the second row of Table 3. We can see that the recommendation performance becomes significant worst. This help confirm our conjecture that the strict ordering of weight on matrix factorization and other regularizations can be an inherent limitation for those approaches.

## 6 CONCLUSION AND DISCUSSION

This work provides a complete analysis on the recently proposed linear models for recommendation systems. Despite that models leverage different deep learning techniques, they achieve similar performance. We find that this is not coincident: all the models add either a nuclear-norm-based (Lemma 1) or a Frobenius-norm-based regularizer. The nuclear-norm-based approach results in estimators that keep $X$'s singular vectors and shrink its singular values in a quite rigid way ( Proposition 2 and Lemma 2), which limit their prediction power. The Frobenius-norm models are more expressive ( Proposition 1) and effective but their estimators are either full-rank or do not have closed-form solutions. To get the best of both nuclear and Frobenius worlds, we propose two low-rank and closed-form estimators ( Section 4) based on carefully generalizing Frobenius-norm-based regularizers. These estimators have competitive performance against linear performance leaders, and thus concisely pack all the benefits obtained by a recent long line of research and abstract out all the computation nuance.

Table 3: Investigating the weight ordering of Matrix Factorization

| Model | ML-20M | | | Netflix | | | MSD | | |
|---|---|---|---|---|---|---|---|---|---|
| | Recall@20 | Recall@50 | nDCG@100 | Recall@20 | Recall@50 | nDCG@100 | Recall@20 | Recall@50 | nDCG@100 |
| MF/LRR weighted | 0.3806 | 0.5175 | 0.4102 | 0.3484 | 0.4320 | 0.3797 | 0.2508 | 0.3390 | 0.3037 |
| MF sorted | 0.3017 | 0.4507 | 0.3361 | 0.2860 | 0.3801 | 0.3265 | 0.2288 | 0.3148 | 0.2802 |

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

# A    RELATED WORK

There have been extensive researches on recommendation (Aggarwal, 2016). Besides the basic user-based and item-based collaborative filtering (Deshpande & Karypis, 2004), the full-rank linear autoencoder approaches include SLIM (Ning & Karypis, 2011), HOLISM  (Christakopoulou & Karypis, 2014), EASE (Steck, 2019), DLAE  (Steck, 2020), whereas low-rank approaches include (Kabbur et al., 2013; Sedhain et al., 2016; Steck, 2020). All the customized recommendation models have been enforcing zero diagonal constraints for generalization purpose, whereas we show an approximate closed-form solution for a two-term Tikhonov regularization without the zero diagonal constraint can be as effective as these models.

Matrix factorization has been been widely studied in practice, partially due to Netflix competition (Koren et al., 2009). Methods like SVD++  (Koren, 2008) and implicit Alternating Least Square (ALS) method (Hu et al., 2008) (also weighted matrix factorization) have been very influential. Jin et al. (2021) shows the relationship between linear autoencoders and matrix factorization, and pointed out a potential advantage of linear autoencoders. In this work, we take a step further to reveal a deeper relationship between Tikhonov regularized linear autoencoders and a few other regularizations including matrix factorization, and show the potential limitation of the class of regularization. We also utilize the variational linear autoencoders (VLAE) to study how the deep VAE based recommendation approaches  (Li & She, 2017; Liang et al., 2018; Shenbin et al., 2020) relate to linear autoencoders and matrix factorization.

Outside recommendation, there have been a few recent studies on regularization landscapes of linear (variational) autoencoders (Kunin et al., 2019; Bao et al., 2020; Lucas et al., 2019a). They do not provide the general weighted $\ell_2$ regularization and thus did not find the inherent limitation on the regularization (for MF). Our VLAE inspired regularization is also never studied before.

Nuclear-norm regularizers can recover low-rank matrices in the vector regression setting (Negahban & Wainwright, 2011). Its weighted generalization can be applied in the area of image processing (Gu et al., 2014). Because weighted nuclear-norm is usually not convex or differentiable, finding optimal solutions is difficult except for a few special cases  (Chen et al., 2013).

# B    APPROXIMATE LOW RANK EDLAE

## B.1    LOW-RANK CLOSED-FORM SOLUTION FOR DLAE

Recall that full-rank DLAE problem:

$$\min_{W} ||X - XW||_F^2 + ||\Lambda^{\frac{1}{2}} W||_F^2 \quad s.t. \quad \Lambda = \frac{p}{1-p} dMat(diag(X^T X)) \tag{17}$$

And now consider the following basic low-rank DLAE model:

$$\arg \min_{rank(W) \leq k} ||X - XW||_F^2 + ||\Lambda^{\frac{1}{2}} W||_F^2 \quad s.t. \quad \Lambda = \frac{p}{1-p} dMat(diag(X^T X)) \tag{18}$$

We leverage the strategy used in (Jin et al., 2021) to derive the low-rank closed-form solution. We formalize the low-rank problem as a standard regression problem:

$$\overline{Y} = \begin{bmatrix} X \\ 0 \end{bmatrix} \quad \overline{X} = \begin{bmatrix} X \\ \Lambda^{\frac{1}{2}} \end{bmatrix} \tag{19}$$

Equiped with this, the above regression problem can be formulated as:

$$\min_{rank(W) \leq k} ||\overline{Y} - \overline{X}W||_F^2$$
$$= \min_{rank(W) \leq k} ||\overline{Y} - \overline{X}W^*||_F^2 + ||\overline{X}W^* - \overline{X}W||_F^2,$$

where $W^* = \arg \min ||\overline{Y} - \overline{X}W^*||_F^2$, is the optimal solution for the full-rank regression problem Eq. (17). The primary loss function $||\overline{Y} - \overline{X}W||_F^2$ then can be decomposed into two parts: $||\overline{Y} - \overline{X}W^*||_F^2$ (full-rank regression problem), and $||\overline{X}W^* - \overline{X}W||_F^2$ (low-rank approximation problem).

Since $W^*$ is the optimum, the vector ($\overline{Y} - \overline{X}W^*$) is orthogonal to $\overline{X}W^* - \overline{X}W = \overline{X}(W^* - W)$ (van Wieringen, 2020), the above equation holds.

Thus the primary low-rank regression problem Eq. (18) can be broken into two subproblems:

**(Subproblem 1:) full-rank closed-from solution:**

$$W^* = \arg\min ||\overline{Y} - \overline{X}W^*||_F^2$$
$$= (X^TX + \Lambda)^{-1}X^TX$$

**(Subproblem 2:) low-rank approximation solution:**

$$\widehat{W} = \arg\min_{rank(W)\leq k} ||\overline{X}W^* - \overline{X}W||_F^2$$
$$= \arg\min_{rank(W)\leq k} ||XW^* - XW||_F^2 + ||\Lambda^{\frac{1}{2}}(W^* - W)||_F^2$$

Denote $\overline{Y}^* = \overline{X}W^*$, the best rank $k$ approximation of $\overline{Y}^*$ in Frobenius norm can be derived from SVD (Eckart & Young, 1936):

$$\overline{Y}^* \overset{\text{SVD}}{=} P\Sigma Q^T$$
$$\overline{Y}^*(k) = P_k\Sigma_k Q_k^T$$

where $Q_k$ takes the first $k$ rows of matrix $Q$, as well as $P$ and $\Sigma$. Since, $Q$ are orthogonal matrices, we notice that:

$$\overline{Y}^*(k) = P_k\Sigma_k Q_k^T = P\Sigma Q^T(Q_kQ_k^T)$$
$$= \overline{X}W^*(Q_kQ_k^T) = \overline{X}W$$

Imediately, we have

$$\boxed{\widehat{W} = W^*(Q_kQ_k^T)}$$

The complete low-rank closed-form solution for Eq. (18) is:

$$\boxed{\widehat{W} = (X^TX + \Lambda)^{-1}X^TX(Q_kQ_k^T)}$$

## B.2 APPROXIMATE LOW-RANK CLOSED-FORM SOLUTION FOR EDLAE (LR-EDLAE-1)

Recall that that low-rank EDLAE aims to solve

$$W = \arg\min_{rank(W)\leq k} ||X - X(W - dMat(diag(W)))||_F^2 + ||\Lambda^{\frac{1}{2}}(W - dMat(diag(W)))||_F^2 \quad (20)$$

When the rank $k$ of the low-rank model is sufficiently large (for instance $k \approx 1000$), its objective function can be approximate as (Steck, 2020):

$$W = \arg\min_{rank(W)\leq k} ||X - XW||_F^2 + ||\Lambda^{\frac{1}{2}}W||_F^2$$
$$s.t. \quad diag(W) = 0 \quad (21)$$

Note that for most of our experimental studies, the rank $k$ in the low-rank model is typically fairly large (over $1K$), and thus the above approximation is reasonable. Compared to Eq. (18), low-rank EDLAE imposes only an additional zero-diagonal constraint. Our goal is to mimic the strategy developed in Appendix B.1 to build a two-staged algorithm to produce a low-rank approximation. We have the following key Lemma.

**Lemma 3.** *Let $W^*$ be a (full rank) solution to*

$$\min_{W^*} \quad ||\overline{Y} - \overline{X}W^*||_F^2$$
$$diag(W^*) = 0.$$

*where,*

$$\overline{Y} = \begin{bmatrix} X \\ 0 \end{bmatrix} \quad \overline{X} = \begin{bmatrix} X \\ \Lambda^{\frac{1}{2}} \end{bmatrix}$$

*Let $W$ be any matrix such that $diag(W) = 0$. We have*

$$\langle \overline{Y} - \overline{X}W^*, \overline{X}W^* - \overline{X}W \rangle = 0.$$

*Proof.* First, observe that $W^*$ can be found by solving a sequence of independent linear regressions, each of which correpsonds to a column of $\overline{Y}$. Let $\overline{X}_{(-1)}$ be the matrix that removes the $i$-th column of $\overline{X}$. Then the $i$-th column of $W^*$, namely $W_i^*$ can be found using two steps:

- *Step 1.* $\hat{W}_i = (\overline{X}_{(-i)}^{\mathrm{T}} \overline{X}_{(-1)})^{-1} \overline{X}_{(-i)}^{\mathrm{T}} Y$.

- *Step 2.* Construct $W_i^*$ by inserting a zero between the $(i-1)$-st and $i$-th entries of $\hat{W}_i$.

This is because enforcing $W_{i,i}^* = 0$ is the same as removing the $i$-th column in $\overline{X}$ to fit $Y_i$.

Next, we move to show that for any $i$,

$$\langle \overline{Y}_i - \overline{X}W_i^*, \overline{X}W_i^* - \overline{X}W_i \rangle = 0.$$

Observing that $W_i^*$ is the optimal solution for $\|\overline{Y}_i - \overline{X}W_i^*\|_2^2$, $\overline{Y}_i - \overline{X}W_i^*$ is orthogonal to the column space of $\overline{X}_{(-i)}$. On the other hand, $\overline{X}W_i^* - \overline{X}W_i = \overline{X}(W_i^* - W_i)$ and $(W_i^* - W_i)_i = 0$. This implies that $\overline{X}W_i^* - \overline{X}W_i$ is in the column space of $\overline{X}_{(-i)}$. Therefore, $\overline{Y}_i - \overline{X}W_i^*$ is orthogonal to $\overline{X}W_i^* - \overline{X}W_i$, which directly implies the lemma. $\qquad \square$

Using Lemma 3, we see that

$$\min_{rank(W) \le k} \|\overline{Y} - \overline{X}W\|_F^2$$
$$s.t. \quad diag(W) = 0$$

$$= \min_{rank(W) \le k} \|\overline{Y} - \overline{X}W^*\|_F^2 + \|\overline{X}W^* - \overline{X}W\|_F^2,$$
$$s.t. \quad diag(W^*) = 0 \quad diag(W) = 0 \tag{22}$$

$$\ge \min_{rank(W) \le k} \|\overline{Y} - \overline{X}W^*\|_F^2 + \|\overline{X}W^* - \overline{X}W\|_F^2,$$
$$s.t. \quad diag(W^*) = 0$$

The last inequality holds because one constraint $diag(W) = 0$ is removed so the objective will only improve. With the above analysis, we are ready to describe our algorithm.

**(Subproblem 1:) full-rank closed-from solution:**

$$W^* = \arg\min \|\overline{Y} - \overline{X}W^*\|_F^2$$
$$= \arg\min_W \|X - XW^*\|_F^2 + \|\Lambda^{\frac{1}{2}}W^*\|_F^2 \tag{23}$$
$$s.t. \quad diag(W^*) = 0$$

The closed-form solution is:

$$C = (X^T X + \Lambda)^{-1}$$
$$W^* = I - C \cdot dMat(1 \oslash diag(C))$$

Table 4: Time complexity analysis for the models listed in Table 1. $X \in \mathbb{R}^{m \times n}$. Assume $m > n$. And we treat $X^T X$ as precomputed.

| Models | Complexity Bounding Operation | Complexity |
|---|---|---|
| 1. Regularized PCA | Singular Value Decomposition | $O(kn^2 + knm)$ |
| 2. MF dropout | Eigen Decomposition | $O(m^2 n + mn^2)$ |
| 4. VLAE | $k$ leading Eigen Decomposition | $O(km^2 + kn^2)$ |
| 5. EASE  6. DLAE  7. EDLAE | Multiplication | $O(n^{2.373})$ |
| 9. EDLAE-ADMM | ADMM | $O(t2^{2.373})$ |
| 3. WLAE  8. LRR  10. LR-DLAE 11. LR-EDLAE-1  12. LR-EDLAE-2 | $k$ leading Eigen Decomposition | $O(kn^2 + n^{2.373})$ |

**(Subproblem 2:) low-rank matrix approximation:**

$$
\begin{aligned}
\widehat{W} &= \arg \min_{rank(W) \leq k} ||\overline{X}W^* - \overline{X}W||_F^2 \\
&= \arg \min_{rank(W) \leq k} ||XW^* - XW||_F^2 + ||\Lambda^{\frac{1}{2}}(W^* - W)||_F^2
\end{aligned}
\tag{24}
$$

Note, in the full-rank subproblem Eq. (23), it forces the diagonal of derived matrix ($W^*$) to be zero. Here, we relax the zero-diagonal constraint - the diagonal of low rank approximate matrix $\widehat{W}$ doesn't have to be strictly zero.

Given this, the closed-from solution (LR-EDLAE-1) of Eq. (24) is given by:

$$
\boxed{\widehat{W} = W^*(Q_k Q_k^T)}
$$

where $Q_k$ comes from: $\overline{X}W^* \overset{\text{SVD}}{=} P\Sigma Q^T$.

**Intuition on the efficacy of the proposed approximation.** We believe that because the full-rank $W^*$ performs well for recommendation accuracy and better than the low-rank solutions (Steck, 2020; Jin et al., 2021), when $W$'s estimation is closer to $W^*$ with respect to the final recovery of $Y$, $XW \approx XW^*$, it has a comparable performance against $XW^*$. In other words, it may not be essential for the diagonal of low-rank approximate matrix $\widehat{W}$ to be zero.

### B.3   SVD OF $W^*$ (LR-EDLAE-2)

Furthermore, if we consider the inequality

$$
\min_{rank(W) \leq k} ||\overline{X}W^* - \overline{X}W||_F^2 \leq \min_{rank(W) \leq k} ||\overline{X}||_F^2 ||W^* - W||_F^2,
$$

or a tighter one

$$
\min_{rank(W) \leq k} ||\overline{X}W^* - \overline{X}W||_F^2 \leq \min_{rank(W) \leq k} ||\overline{X}||_2^2 ||W^* - W||_F^2
$$

Both suggest if we simply perform SVD on $W^*$, we can obtain an upper bound error between $\overline{X}W^*$ and $\overline{X}W$.

Simply, the other closed-form solution $\widehat{W}$ (LR-EDLAE-2) is:

$$
W^* \overset{\text{SVD}}{=} U\Sigma V^T
$$
$$
\widehat{W} = U_k \Sigma_k V_k^T
$$

## C   TIME COMPLEXITY ANALYSIS

The running time of all linear models discussed in this paper is determined by one or more of the following factors: *(i) Computation of matrix multiplication and inversion.* The best known theoretical bound is $O(n^{2.373})$(Alman & Williams, 2021) but to the authors' knowledge the algorithm is not implemented in any mainstream libraries. *(ii) Computation of $k$ leading singular vectors.* The running

time is $O(kn^2)$ for an $n \times n$ matrix. But recent trends believe that the constant terms in the big-$O$ notation matter so the asymptotic analysis becomes less fashionable (Stathopoulos & McCombs, 2010; Wu et al., 2017). *(iii) Convergence rate of a gradient-based algorithm.* Except for special cases, when an objective is not convex (e.g., when $W$ is decomposed into $W = PQ$), the convergence rate of an algorithm is remarkably difficult to analyze and often has substantial discrepancies with its practical performance (Bhojanapalli et al., 2016).

We remark that all three factors exhibit significant gaps between theory and practice, i.e., the best theoretical algorithms are not implemented, constants in big-O matter, and accurate characterization of convergence rate is difficult. Therefore:

In theory, *(i)* the running time for full-rank Frobenius-norm models (cases 5. EASE, 6. DLAE, and 7. EDLAE in Table 1) is $O(n^{2.373})$ because they involve merely matrix inversion and multiplication operations; *(ii)* the running time for most low-rank models (cases 3. WLAE, 8. LRR, 10. LR-DLAE, 11. LR-EDLAE-1, and 12. LR-EDLAE-2 in Table 1) are $O(n^{2.373} + kn^2)$ because they require computation of $k$ leading singular vectors or eigenvectors; *(iii)* the nuclear-norm models require the multiplication with $X \in \mathbb{R}^{m \times n}$. So the running time for cases 1 (Regularized PCA), 2 (MF dropout), and 4 (VLAE) are $O(kn^2 + mnk), O(m^2n + mn^2), O(km^2 + kn^2)$ respectively; and *(iv)* The running time ADMM-based model (case 9. EDLAE-ADMM in Table 1) is $O(tn^{2.373})$, where $t$ is the number of iteration needed to converge. We summarize the results in Table 4

In practice, solving low-rank SVD and computing matrix multiplication require similar time especially after carefully using HPC techniques. So our proposed algorithms LR-EDLAE-1 and LR-EDLAE-2 (Section 4) are neither slower nor faster than existing ones that do not use gradient algorithms (e.g. full-rank EDLAE, etc.). The clock time for gradient-based (e.g. ADMM) algorithms is usually worse than that in theory because determination of hyper-parameters such as learning rate is usually a guesswork. A rule of thumb is that LR-EDLAE-1 and LR-EDLAE-2 are faster for small $k$ and slower (than e.g., ADMM-based methods) for large $k$.

Finally, we would like to address again on the advantage of having any closed-form solutions: they are easy to implement and use, and can be quickly tested using any standard matrix computation platform without the specialized recommendation library. They are more robust to error and bug, and can serve as the competitive baseline for recommendation evaluation tasks.

# D  PROOFS

## D.1  PROOF OF LEMMA 1

Consider $X \in \mathbb{R}^{m \times n}$. Suppose latent variables $z \in \mathbb{R}^k$ and generate data $x \in \mathbb{R}^n$. The Variational Linear Autoencoder (VLAE) is defined in the same way as (Lucas et al., 2019b):

$$
\begin{aligned}
p(x \mid z) &= \mathcal{N}\left(Wz + \boldsymbol{\mu}, \sigma^2 I\right) \\
q(z \mid x) &= \mathcal{N}(V(x - \boldsymbol{\mu}), D)
\end{aligned}
\tag{25}
$$

For simplification, we set $\mu = 0$ in following context. And the ELBO of VLAE is known as:

$$
\begin{aligned}
\mathcal{L}_x = & -KL(q(z|x)||p(z)) + \mathbb{E}_{q(z|x)}[\log p(x|z)] \\
KL(q(z|x)||p(z)) = & -\log|D| + x^T V^T V x + tr(D) - k \\
\mathbb{E}_{q(z|x)}[\log p(x|z)] = & -\frac{1}{2\sigma^2}\Big(tr(WDW^T) + x^T V^T W^T W V x - 2x^T W V x + x^T x\Big) \\
& -\frac{n}{2}\log 2\pi\sigma^2
\end{aligned}
\tag{26}
$$

Again, the (maximizing) ELBO can be written as:

$$\mathcal{L}_x = -\frac{1}{2}\Big(-\log|D| + x^T V^T V x + tr(D) - k\Big) - \frac{n}{2}\log 2\pi\sigma^2$$
$$- \frac{1}{2\sigma^2}\Big(tr(WDW^T) + x^T V^T W^T W V x - 2x^T W V x + x^T x\Big) \tag{27}$$
$$= -\frac{1}{2}||Vx||_2^2 - \frac{1}{2\sigma^2}\Big(||W\sqrt{D}||_F^2 + ||x - WVx||_2^2\Big) + f(D,\sigma)$$

where $f(D,\sigma) = \frac{1}{2}\log|D| - \frac{1}{2}tr(D) + \frac{k}{2} - \frac{n}{2}\log 2\pi\sigma^2$, $x \in \mathbb{R}^n$ and $z \in \mathbb{R}^k$.

For whole data, it is equivalent to minimize:

$$\mathcal{L} = ||X^T - WVX^T||_F^2 + m||W\sqrt{D}||_F^2 + \sigma^2||VX^T||_F^2 + g(D,\sigma)$$
$$= ||X - XV^T W^T||_F^2 + m||\sqrt{D}W^T||_F^2 + \sigma^2||XV^T||_F^2 + g(D,\sigma) \tag{28}$$

where $g(D,\sigma) = -\sigma^2 m\big(\log|D| - tr(D) + k - n\log 2\pi\sigma^2\big)$.

## D.2 PROOF OF PROPOSITION 1

Note that when $\sigma_i \le \lambda_{(k-i)}$, the new singular value shrinks to zero, and can be removed. Basically, for any $\lambda_{(1)} \ge \cdots \ge \lambda_{(k)}$, we can build the corresponding Tikhonov regularized instance by setting

$$\frac{\sigma_i^2}{\sigma_i^2 + \lambda_i'} = \frac{\sigma_i - \lambda_{(k-i)}}{\sigma_i}, \text{i.e.}, \lambda_k' = \frac{\sigma_i^3}{\sigma_i - \lambda_{(k-i)}} - \sigma_i^2. \tag{29}$$

**Discussion of Proposition 1:** Further, the same observation holds true for the regularization (Eq. (2)), and the weighted-nuclear norm regularization in when the weights are in the non-ascending order. This observation suggests a potentially limitation of the earlier regularization as they will always try to maintain the larger singular values: when a singular value is large, the shrinkage will be small. Such regularization has shown to work well in the areas such as image processing (Gu et al., 2014). But it has not been studied or confirmed if it will work for the recommendation. In Section 5, we report our experimental study which shows such regularization could be too restrictive for recommendation.

## D.3 "CLEAR UP" EQ. (4)

Since our objective is for recommendation (not purely on recovering the low rank factors of the data), we treat the covariance matrix $D$ as hyperparameters. Thus, the term $g(D,\sigma)$ becomes a constant, and let $A = \sigma V^T$, $B = 1/\sigma W^T$, $\Lambda = \sigma\sqrt{mD}$.

If we consider $D$ as an optimization parameter, then the optimal solution of VLAE is equivalent to that of pPCA (Tipping & Bishop, 1999a). In this case, $D = \sigma^2(\Sigma^2/m)^{-1}$, where $\Sigma^2 = diag(\sigma_i^2)$ are the eigen-values of the covariance matrix of $X$, and $\sigma^2 = \frac{1}{n-k}\sum_{j=k+1}^n \frac{\sigma_j^2}{m}$. Further, the closed-form solutions of $V$ ($A$) and $W$ ($B$) are characterized. However, for recommendation, the matrix $D$ can be considered as a hyperparameter (to be tuned); in this case, the closed-form solution is not studied yet.

It is easy to see we can "clear up" Eq. (4) and obtain the following optimization problem :

$$\min_{A \in \mathbf{R}^{n \times k}, B \in \mathbf{R}^{k \times n}} ||X - XAB||_F^2 + ||XA||_F^2 + ||\Lambda B||_F^2, \tag{30}$$

in which decision variables are $A$ and $B$, and the hyperparameter is a diagonal matrix $\Lambda \in \mathbf{R}^{k \times k}$.

## D.4 INTUITION FOR PROVING PROPOSITION 2

Here, We explain the intuition for proving Appendix D.5 Proposition 2 (see Appendix D.5 for the full analysis). Consider $OPT1$ and let $W = PQ$. Our goal is to characterize the behaviors of $P$ and $Q$ with the presence of the regularizers when $W = PQ$ is known (fixed). Let the SVD of $W$ be $U_W \Sigma_W V_W^T$. Because two regularizers $||\Lambda^{\frac{1}{2}}Q||_F^2$ and $||P\Lambda^{\frac{1}{2}}||_F^2$ are symmetric, we could "guess" $P = U_W \Sigma_W^{\frac{1}{2}}\Omega$ and $Q = \Omega^T \Sigma_W^{\frac{1}{2}} V_W^T$, where $\Omega$ is a unitary matrix. Now we have

$$\|\Lambda^{\frac{1}{2}}Q\|_F^2 + \|P\Lambda^{\frac{1}{2}}\|_F^2 = \|\Lambda^{\frac{1}{2}}\Omega\Sigma^{\frac{1}{2}}V_W^T\|_F^2 + \|U_W\Sigma_W^{\frac{1}{2}}\Omega\Lambda^{\frac{1}{2}}\|_F^2 = 2\|\Lambda^{\frac{1}{2}}\Omega\Sigma_W^{\frac{1}{2}}\|_F^2.$$

Now the question of finding $P$ and $Q$ when $W$ is known boils down to finding a unitary matrix $\Omega$ that minimizes $\|\Lambda^{\frac{1}{2}}\Omega\Sigma_W^{\frac{2}{2}}\|_F^2$, where diagonal matrices $\Lambda$ and $\Sigma_W$ are given. Recall that $\lambda_i = \Lambda_{ii}$ and let $\sigma_i = (\Sigma_W)_{ii}$. Note that $\lambda_i$'s could be unsorted and, and that $\sigma_i$'s are sorted in descending order.

If we restrict $\Omega$ to be only a permutation matrix, then we aim to find a permutation $\pi \in [k]$ that minimizes $\sum_{i \leq k} \lambda_{\pi(i)}\sigma_i$. Using a rearrangement inequality (Yue, 2020), we can see that the minimal is achieved when $\lambda_{\pi(1)} \leq \lambda_{\pi(2)} \leq \cdots \leq \lambda_{\pi(k)}$. In this case, we indeed have $\min_{\Omega \text{ a permutation}} \|\Lambda^{\frac{1}{2}}\Omega\Sigma_W^{\frac{2}{2}}\|_F^2 = \|\Sigma_W\|_{\omega,*}$, where $\omega = (\lambda_{\pi(1)}, \ldots, \lambda_{\pi(k)})$.

Note that because $PQ = P\Omega\Omega^T Q$ for any unitary matrix $\Omega$, it is always beneficial to use $\Omega$ to shuffle the rows and columns of $P$ and $Q$ so that the largest $\sigma_i$ is mapped to the smallest $\lambda_i$, etc. This "degree of freedom" from $\Omega$ also explains why ordering the values along $\Lambda$'s diagonal is irrelevant.

Appendix D.5 shows that even when $\Omega$ is allowed to be any unitary matrix, the optimal one is still a permutation matrix. This conclusion can be viewed as a matrix version of re-arrangement inequality.

### D.5 PROOF OF PROPOSITION 2

By slightly abusing the notation, we shall let $OPT1$ ($OPT2$) be the value of the optimal solution for $OPT1$ ($OPT2$). We need to show that $OPT1 = OPT2$. We need two directions.

$OPT2 \geq OPT1$: Let $W^*$ be an optimal solution for $OPT2$. Let the SVD of $W^*$ be $U^*\Sigma^*(V^*)^T$. Recall that $W^*$ needs to satisfy the rank constraint $\text{rank}(W^*) \leq k$ so $U^* \in \mathbf{R}^{m \times k}$, $\Sigma^* \in \mathbf{R}^{k \times k}$, and $V^* \in \mathbf{R}^{n \times k}$. Let $\pi$ be a permutation on $[k]$ such that $\lambda_{\pi(1)} \leq \lambda_{\pi(2)} \leq \cdots \leq \lambda_{\pi(k)}$. Let also $\Omega$ be the corresponding permutation matrix. Specifically, $\Omega \in \{0,1\}^{k \times k}$ and there is exactly one entry in each row of $\Omega$ is 1:

$$\Omega_{i,j} = \begin{cases} 1 & \text{if } j = \pi(i). \\ 0 & \text{otherwise.} \end{cases}$$

For example, consider a case in which $\lambda_1 > \lambda_2 > \cdots > \lambda_k$. Then we set $\pi = (k, k-1, \ldots, 1)$, and correspondingly,

$$\Omega = \begin{pmatrix} 0 & \ldots & 0 & 1 \\ 0 & \ldots & 1 & 0 \\ & \ldots & & \\ 1 & \ldots & 0 & 0 \end{pmatrix}.$$

Next, let $P = U^*(\Sigma^*)^{\frac{1}{2}}\Omega$ and $Q = \Omega^T(\Sigma^*)^{\frac{1}{2}}(V^*)^T$. We have $W^* = PQ$ and $f(W^*) = f(PQ)$. In addition,

$$\|\Lambda^{\frac{1}{2}}Q\|_F^2 + \|P\Lambda^{\frac{1}{2}}\|_F^2 = 2\|\Lambda^{\frac{1}{2}}\Omega(\Sigma^*)^{\frac{1}{2}}\|_F^2 = 2\sum_{i \leq k}\lambda_{\pi(i)}\sigma_i = 2\|W^*\|_{\omega,*},$$

where $\sigma_i$ is the $i$-th largest singular value of $W^*$. In other words, we have found a $(P,Q)$ pair such that

$$f(PQ) + \|\Lambda^{\frac{1}{2}}Q\|_F^2 + \|P\Lambda^{\frac{1}{2}}\|_F^2 = f(W^*) + 2\|W^*\|_{\omega,*} = OPT2,$$

which shows that $OPT1 \leq OPT2$.

$OPT2 \leq OPT1$. Let $P^*$ and $Q^*$ be an optimal solution for $OPT1$. Let the singular values of $P^*$ be $\sigma_1(P^*) \geq \sigma_2(P^*) \geq \cdots \geq \sigma_k(P^*)$ and those of $Q^*$ be $\sigma_1(Q^*) \geq \sigma_2(Q^*) \geq \cdots \geq \sigma_k(Q^*)$. Let also $\sigma_1^* \geq \cdots \geq \sigma_k^*$ be the singular values of $P^*Q^*$.

We shall find a lower bound of $\|\Lambda^{\frac{1}{2}}Q\|_F^2 + \|P\Lambda^{\frac{1}{2}}\|_F^2$ expressed in terms of $\sigma_i^*$'s. In fact, we shall show that

$$\|\Lambda^{\frac{1}{2}}Q\|_F^2 + \|P\Lambda^{\frac{1}{2}}\|_F^2 \geq 2\|P^*Q^*\|_{\omega,*}. \tag{31}$$

One can see that if Eq. (31) were true, we have

$$OPT2 \leq f(P^*Q^*) + 2\|P^*Q^*\|_{\omega,*} \leq f(P^*Q^*) + \|\Lambda^{\frac{1}{2}}Q^*\|_F^2 + \|P^*\Lambda^{\frac{1}{2}}\|_F^2 = OPT1.$$

Thus, it remains to prove Eq. (31). Let $\lambda_{(1)} \geq \lambda_{(2)} \geq \cdots \geq \lambda_{(k)}$ be a sorted sequence of $\lambda_i$'s i.e., $\lambda_{(k)} = \lambda_{\pi(1)}, \lambda_{(k-1)} = \lambda_{\pi(2)}, \ldots, \lambda_{(1)} = \lambda_{\pi(k)}$.

First, we show that $\|P\Lambda^{\frac{1}{2}}\|_F^2 \geq \sum_{i=1}^{k} \lambda_{(k-i+1)} \times \sigma_i^2(P^*)$ and $\|\Lambda^{\frac{1}{2}}Q\|_F^2 \geq \sum_{i=1}^{k} \lambda_{(k-i+1)} \times \sigma_i^2(Q^*)$. We need the following Lemma (see e.g., Theorem 2 in (Yue, 2020)):

**Lemma 4.** *Let $A$ and $B$ be two positive definite matrices in $\mathbf{R}^{k \times k}$. Then it holds that*

$$\sum_{i=1}^{k} \sigma_i(A)\sigma_{k-i+1}(B) \leq \operatorname{tr}(B^{\frac{1}{2}}AB^{\frac{1}{2}}). \tag{32}$$

Let the SVD of $P^*$ be $U_{P^*}\Sigma_{P^*}V_{P^*}^{\mathrm{T}}$ and that of $Q^*$ be $U_{Q^*}\Sigma_{Q^*}V_{Q^*}^{\mathrm{T}}$. We have

$$\|P^*\Lambda^{\frac{1}{2}}\|_F^2 = \|U_{P^*}\Sigma_{P^*}V_{P^*}^{\mathrm{T}}\Lambda^{\frac{1}{2}}\|_F^2 = \|\Sigma_{P^*}V_{P^*}^{\mathrm{T}}\Lambda^{\frac{1}{2}}\|_F^2 = \operatorname{tr}(\Sigma_{P^*}V_{P^*}^{\mathrm{T}}\Lambda V_{P^*}\Sigma_{P^*}). \tag{33}$$

We now apply Lemma 4 by setting $A = V_{P^*}^{\mathrm{T}}\Lambda V_{P^*}$ and $B = \Sigma_{P^*}^2$, and obtain that

$$\|P^*\Lambda^{\frac{1}{2}}\|_F^2 = \operatorname{tr}(\Sigma_{P^*}V_{P^*}^{\mathrm{T}}\Lambda V_{P^*}\Sigma_{P^*}) \geq \sum_{i=1}^{k} \lambda_{(k+1-i)} \times \sigma_i^2(P^*). \tag{34}$$

We may similarly prove that $\|\Lambda^{\frac{1}{2}}Q^*\|_F^2 \geq \sum_{i=1}^{k} \lambda_{(k-i+1)} \times \sigma_i^2(Q^*)$. Therefore,

$$\|\Lambda^{\frac{1}{2}}Q^*\|_F^2 + \|P^*\Lambda^{\frac{1}{2}}\|_F^2 \geq \sum_{i=1}^{k} \lambda_{(k+1-i)} \times (\sigma_i^2(P^*) + \sigma_i^2(Q^*)) \tag{35}$$

provides a lower bound of $\|\Lambda^{\frac{1}{2}}Q^*\|_F^2 + \|P^*\Lambda^{\frac{1}{2}}\|_F^2$ in terms of $\sigma_i(P^*)$ and $\sigma_i(Q^*)$. We next aim to express the lower bound in terms of $\sigma_i^*$'s (singular values of $P^*Q^*$) directly.

The following program gives a lower bound for $\|\Lambda^{\frac{1}{2}}Q^*\|_F^2 + \|P^*\Lambda^{\frac{1}{2}}\|_F^2$:

$$\begin{aligned} \min: \quad & \|\Lambda^{\frac{1}{2}}Q^*\|_F^2 + \|P^*\Lambda^{\frac{1}{2}}\|_F^2 \\ \text{subject to} \quad & W = P^*Q^* \\ & \sigma_i(W) = \sigma_i^* \quad \text{for } i \leq k. \end{aligned} \tag{36}$$

Write the SVD of $W$ be $U_W\Sigma_W V_W^{\mathrm{T}}$. Also, let $\tilde{P} = U_W^{\mathrm{T}}P^*$ and $\tilde{Q} = Q^*V_W$. Noting that the columns in $P^*$ are in the column space of $W$ and the rows in $Q^*$ are in the row space of $W$, we have *(i)* $\sigma_i(P^*) = \sigma_i(\tilde{P})$ and $\sigma_i(Q^*) = \sigma_i(\tilde{Q})$ for $i \leq k$, and *(ii)* $\|\Lambda^{\frac{1}{2}}Q^*\|_F^2 + \|P^*\Lambda^{\frac{1}{2}}\|_F^2 = \|\Lambda^{\frac{1}{2}}\tilde{Q}\|_F^2 + \|\tilde{P}\Lambda^{\frac{1}{2}}\|_F^2$.

Therefore, Eq. (36) can be equivalently written as

$$\begin{aligned} \min: \quad & \|\Lambda^{\frac{1}{2}}\tilde{Q}\|_F^2 + \|\tilde{P}\Lambda^{\frac{1}{2}}\|_F^2 \\ \text{subject to} \quad & \Sigma_W = \tilde{P}\tilde{Q} \\ & (\Sigma_W)_{i,i} = \sigma_i^* \quad \text{for } i \leq k. \end{aligned} \tag{37}$$

Now $\tilde{P}\tilde{Q}$ is positive definite. Using a similar technique developed in (Bao et al., 2020) (Theorem 1), one can see that $\tilde{P} = \tilde{Q}^{\mathrm{T}}$. See also Lemma 5. This implies that $\tilde{P} = \Sigma_W^{\frac{1}{2}}\Omega$ for some unitary matrix $\Omega$ and $\sigma_i(P^*) = \sigma_i(\tilde{P}) = \sigma_i(Q^*) = \sigma_i(\tilde{Q}) = \sqrt{\sigma_i^*}$ for $i \leq k$. Together with Eq. (35), we have

$$\|\Lambda^{\frac{1}{2}}Q^*\|_F^2 + \|P^*\Lambda^{\frac{1}{2}}\|_F^2 \geq 2\sum_{i \leq k} \lambda_{(k+1-i)} \times \sigma_i^2(P^*) = 2\sum_{i \leq k} \lambda_{(k+1-i)} \times \sigma_i^* = 2\|P^*Q^*\|_{\omega,*}.$$

### D.6 PROOF OF COROLLARY 1

We first find an optimal solution for Eq. (9). Let the SVD of $X$ be $X = U_X \Sigma_X V_X^{\mathrm{T}}$, where $U_X \in \mathbf{R}^{m \times n}$, $\Sigma_X \in \mathbf{R}^{n \times n}$, and $V_X \in \mathbf{R}^{n \times n}$. Let $\bar{U}_X$ be an arbitrary basis for the subspace that is orthogonal to $X$'s column space so $\bar{U}_X \in \mathbf{R}^{m \times (m-n)}$ and $[U_X, \bar{U}_X]$ form a basis for $\mathbf{R}^m$. We have

$$\|X - PQ\|_F^2 + \|P\Lambda^{\frac{1}{2}}\|_F^2 + \|\Lambda^{\frac{1}{2}}Q\|_F^2$$

$$= \left\| \begin{pmatrix} U_X^{\mathrm{T}} \\ \bar{U}_X^{\mathrm{T}} \end{pmatrix} X V_X - \begin{pmatrix} U_X^{\mathrm{T}} \\ \bar{U}_X^{\mathrm{T}} \end{pmatrix} PQV_X \right\|_F^2 + \left\| \begin{pmatrix} U_X^{\mathrm{T}} \\ \bar{U}_X^{\mathrm{T}} \end{pmatrix} P\Lambda^{\frac{1}{2}} \right\|_F^2 + \|\Lambda^{\frac{1}{2}}QV_X\|_F^2.$$

Let $\tilde{P} = \begin{pmatrix} U_X^{\mathrm{T}} \\ \bar{U}_X^{\mathrm{T}} \end{pmatrix} P$ and $\tilde{Q} = QV_X$. Then our objective becomes

$$\min_{\tilde{P}, \tilde{Q}} \left\| \begin{pmatrix} \Sigma_X \\ 0_{(m-n) \times n} \end{pmatrix} - \tilde{P}\tilde{Q} \right\|_F^2 + \|\tilde{P}\Lambda^{\frac{1}{2}}\|_F^2 + \|\Lambda^{\frac{1}{2}}\tilde{Q}\|_F^2. \tag{38}$$

Let $\tilde{W} = \tilde{P}\tilde{Q}$ and the singular values of $\tilde{W}$ be $\sigma_1^* \geq \sigma_2^* \geq \cdots \geq \sigma_k^*$. Let also $\tilde{\Sigma} = \begin{pmatrix} \Sigma_X \\ 0_{(m-n) \times n} \end{pmatrix}$.

Recall also that $\sigma_i$ is the $i$-th largest singular value of $X$. We next show that

$$\left\| \begin{pmatrix} \Sigma_X \\ 0 \end{pmatrix} - \tilde{P}\tilde{Q} \right\|_F^2 = \|\tilde{\Sigma} - \tilde{P}\tilde{Q}\|_F^2 \geq \sum_{i=1}^k (\sigma_i - \sigma_i^*)^2 + \sum_{i=k+1}^n \sigma_i^2.$$

Note first that

$$\|\tilde{\Sigma} - \tilde{W}\|_F^2 = \|\tilde{\Sigma}\|_F^2 + \|\tilde{W}\|_F^2 - 2\langle \tilde{\Sigma}, \tilde{W} \rangle. \tag{39}$$

Next, we have ((Zheng et al., 2018)):

$$|\langle \tilde{\Sigma}, \tilde{W} \rangle| = |\mathrm{tr}(\tilde{\Sigma}\tilde{W}^{\mathrm{T}})\| \leq |\mathrm{tr}(\tilde{\Sigma}\Sigma_{\tilde{W}})| = \sum_{i=1}^k \sigma_i \sigma_i^*.$$

Therefore, $\langle \tilde{\Sigma}, \tilde{W} \rangle$ is maximized when

$$\tilde{W}_{i,j} = \begin{cases} \sigma_i^* & \text{if } i = j \leq k \\ 0 & \text{Otherwise.} \end{cases}$$

When we plug in this optimized $\tilde{W}$ to Eq. (39), we get

$$\|\tilde{\Sigma} - \tilde{W}\|_F^2 \geq \sum_{i=1}^k (\sigma_i - \sigma_i^*)^2 + \sum_{i=k+1}^n \sigma_i^2.$$

Next, from Proposition 2, we have

$$\|\tilde{P}V^{\frac{1}{2}}\|_F^2 + \|\Lambda^{\frac{1}{2}}\tilde{Q}\|_F^2 \geq \sum_{i=1}^k \lambda_{(k-i+1)}\sigma_i^*.$$

Therefore, we can find a lower bound for Eq. (5) in terms of $\sigma_i^*$'s:

$$\mathcal{L}(\sigma_1^*, \ldots, \sigma_k^*) = \sum_{i=1}^k (\sigma_i - \sigma_i^*)^2 + 2\sum_{i=1}^k \lambda_{(k-i+1)}\sigma_i^* + \sum_{i=k+1}^m \sigma_i^2 \quad (\sigma_1^* \geq \cdots \geq \sigma_k^* \geq 0). \tag{40}$$

We next find a minimal value of $\mathcal{L}$ (by treating $\sigma_i^*$'s as decision variables). This will give us a lower bound (and is independent of $\sigma_i^*$) on our optimization problem. We then show that this lower bound can be achieved by carefully constructing $\tilde{W}$ (as well as $\tilde{P}$ and $\tilde{Q}$). This means such $\tilde{W}$ is optimal.

Specifically, we need to find an optimal solution for the following program:

$$\text{minimize}_{\sigma_1^*,\ldots,\sigma_k^*} \quad \mathcal{L}(\sigma_1^*,\ldots,\sigma_k^*) \tag{41}$$
$$\text{subject to:} \quad \sigma_i^* \geq 0$$
$$\sigma_1^* \leq \sigma_2^* \leq \cdots \leq \sigma_k^* \quad \text{(Ordering constraint)}$$

We shall first find an optimal solution for

$$\text{minimize}_{\sigma_1^*,\ldots,\sigma_k^*} \quad \mathcal{L}(\sigma_1^*,\ldots,\sigma_k^*) \tag{42}$$
$$\text{subject to:} \quad \sigma_i^* \geq 0$$

Note here, the ordering constraint is removed so the optimal value for Eq. (42) should be no more than that for Eq. (41). We shall see that the optimal solution for Eq. (41) also satisfies the ordering constraint so indeed optimal solutions for Eq. (41) and Eq. (42) are the same.

The problem Eq. (42) boils down to finding

$$\min_{\sigma_i^* \geq 0} (\sigma_i - \sigma_i^*)^2 + 2 \sum_{i=1}^{k} \lambda_{(k-i+1)} \sigma_i^*.$$

We note that $\sigma_i^*$'s do not interact with each other so we can optimize each $\sigma_i^*$'s independently. We get

$$\sigma_i^* = (\sigma_i - \lambda_{(k-i+1)})^+.$$

We can check that $\sigma_1^* \geq \cdots \geq \sigma_k^*$. Therefore, the optimal value for Eq. (41) is

$$\sum_{i=1}^{k} (\sigma_i - (\sigma_i - \lambda_{(k-i+1)})^+)^2 + 2 \sum_{i=1}^{k} \lambda_{(k-i+1)} (\sigma_i - \lambda_{(k-i+1)})^+ + \sum_{i=k+1}^{n} \sigma_i^2.$$

This is also a lower bound for Eq. (9). One can check that when we set $P$ and $Q$ as

$$P^* = U_k diag(\sqrt{(\sigma_1 - \lambda_{(k)})^+}, \ldots, \sqrt{(\sigma_k - \lambda_{(1)})^+})\Omega, \tag{43}$$
$$Q^* = \Omega^{\mathrm{T}} diag(\sqrt{(\sigma_1 - \lambda_{(k)})^+}, \ldots, \sqrt{(\sigma_k - \lambda_{(1)})^+})V_k^{\mathrm{T}},$$

the lower bound is achieved so Eq. (43) gives an optimal solution. Here, $U_k$ and $V_k$ are leading left and right singular vectors of $X$.

Now we move to analyze Eq. (10). Our goal is to reduce Eq. (10) to Eq. (9). Let

$$P = XA\Lambda^{-\frac{1}{2}} \qquad Q = \Lambda^{\frac{1}{2}} B.$$

Then Eq. (10) becomes

$$\text{minimize}_{P,Q} \quad \|X - PQ\|_F^2 + \|P\Lambda^{\frac{1}{2}}\|_F^2 + \|\Lambda^{\frac{1}{2}} Q\|_F^2 \tag{44}$$
$$\text{subject to} \quad P = XA\Lambda^{-\frac{1}{2}} \quad \text{(Constraint P)}$$
$$Q = \Lambda^{\frac{1}{2}} B \quad \text{(Constraint Q)}.$$

Here, $X$ and $\Lambda$ are given, whereas $P$, $Q$, $A$, and $B$ are decision variables. The (Constraint P) says that each column of $P$ needs to be in a column space of $X$ (it is a necessary and sufficient condition for $A$ to exist). The (Constraint Q) simply says $Q$ and $B$ are linearly related and does not have tangible impact to the optimization problem.

But we note that when we put aside the constraints, an optimal $(P, Q)$ is specified by Eq. (43). The columns of the optimal $P$ indeed is in the column space of $X$. So $(P, Q)$ is also an optimal solution for Eq. (44). We may find the corresponding $A$ and $B$:

$$A^* = X^{\dagger} P^* \Lambda^{\frac{1}{2}} \quad \text{and} \quad B^* = \Lambda^{-\frac{1}{2}} Q^*,$$

### D.7 SYMMETRIC LEMMA

**Lemma 5.** *Let $\tilde{P}, \tilde{Q} \in \mathbf{R}^{k \times k}$ be full rank, $\Lambda$ be a diagonal matrix, and $\Sigma_W$ be a diagonal matrix so that $(\Sigma_W)_{i,i} = \sigma_i^*$, where $\sigma_i^*$'s are sorted in descending order. Consider the optimization problem:*

$$
\begin{aligned}
\min : \quad & \|\Lambda^{\frac{1}{2}} \tilde{Q}\|_F^2 + \|\tilde{P} \Lambda^{\frac{1}{2}}\|_F^2 && (45) \\
\text{subject to} \quad & \Sigma_W = \tilde{P}\tilde{Q} \\
& (\Sigma_W)_{i,i} = \sigma_i^* \quad \text{for } i \le k.
\end{aligned}
$$

*There is an optimal solution such that $\tilde{P} = \tilde{Q}^{\mathrm{T}}$*

*Proof.* Let $\hat{P} = \tilde{P}\Lambda^{\frac{1}{2}}$ and $\hat{Q} = \Lambda^{-\frac{1}{2}}\tilde{Q}$. The problem Eq. (45) is equivalent to

$$
\begin{aligned}
\min : \quad & \|\Lambda\hat{Q}\|_F^2 + \|\hat{P}\|_F^2 && (46) \\
\text{subject to} \quad & \Sigma_W = \hat{P}\hat{Q} \\
& (\Sigma_W)_{i,i} = \sigma_i^* \quad \text{for } i \le k.
\end{aligned}
$$

Let the SVD of $\hat{Q}$ be $U_{\hat{Q}}\Sigma_{\hat{Q}}V_{\hat{Q}}^{\mathrm{T}}$ so $\hat{Q}^{-1} = V_{\hat{Q}}\Sigma_{\hat{Q}}^{-1}U_{\hat{Q}}^{\mathrm{T}}$. We can also see that $\hat{P} = \Sigma_W\hat{Q}^{-1}$. Therefore, the objective term becomes

$$
\|\Lambda U_{\hat{Q}}\Sigma_{\hat{Q}}V_{\hat{Q}}^{\mathrm{T}}\|_F^2 + \|\Sigma_W V_{\hat{Q}}\Sigma_{\hat{Q}}^{-1}U_{\hat{Q}}^{\mathrm{T}}\|_F^2 = \|\Lambda U_{\hat{Q}}\Sigma_{\hat{Q}}\|_F^2 + \|\Sigma_W V_{\hat{Q}}\Sigma_{\hat{Q}}^{-1}\|_F^2.
$$

Let us consider the stationary points $U_{\hat{Q}}$ and $V_{\hat{Q}}$ when $\Sigma_{\hat{Q}}$ is fixed. We can see that they need to be permutation matrices to minimize both terms in the objective (using the rearrangement inequality again). Therefore, we can see $\hat{Q} = \Sigma_1\Sigma_{\hat{Q}}\Sigma_2$ for two permutation matrices $\Sigma_1$ and $\Sigma_2$. This implies that $\tilde{Q} = \Lambda^{\frac{1}{2}}\Sigma_1\Sigma_{\hat{Q}}\Sigma_2$, i.e., each row (column) of $\tilde{Q}$ has exactly one non-zero entry. We may similarly show that each row (column) of $\tilde{P}$ has exactly one non-zero entry. In addition, the locations of non-zero entries of $\tilde{P}$ and $\tilde{Q}^{\mathrm{T}}$ are identical because $\tilde{P}\tilde{Q}$ is a diagonal matrix. We may thus write

$$
\tilde{P} = \Sigma_{(1)}\Sigma_{\tilde{P}}\Sigma_{(2)} \quad \tilde{Q} = \Sigma_{(2)}^{\mathrm{T}}\Sigma_{(\tilde{Q})}\Sigma_{(1)}^{\mathrm{T}},
$$

where $(\Sigma_{\tilde{P}})_{i,i} = \sigma_i(\tilde{P})$ and $(\Sigma_{(\tilde{Q})})_{i,i} = \sigma_{\tau(i)}(\tilde{Q})$, where $\tau$ is a permutation on $[k]$. The set of (possibly unsorted) singular values for $\tilde{P}\tilde{Q}$ thus is $\sigma_i(\tilde{P})\sigma_{\tau(i)}(\tilde{Q})$. Thus, we can see that there exists a permutation $\bar{\pi}$ such that

$$
\begin{aligned}
\|\Lambda^{\frac{1}{2}}\tilde{Q}\|_F^2 + \|\tilde{P}\Lambda^{\frac{1}{2}}\|_F^2 &= \sum_{i \le k} \left( \sigma_i^2(P^*)\lambda_{\bar{\pi}(i)} + \sigma_{\tau(i)}^2(Q^*)\lambda_{\bar{\pi}(i)} \right) \\
&\ge \sum_{i \le k} 2\sigma_i(P^*)\sigma_{\tau(i)}^*(Q)\lambda_{\bar{\pi}(i)} \\
&\ge 2\|P^*Q^*\|_{\omega,*}.
\end{aligned}
$$

One can see that we can set $\tilde{P} = \tilde{Q}^{\mathrm{T}}$ to make all inequality becomes equality so there is an optimal solution such that $\tilde{P} = \tilde{Q}^{\mathrm{T}}$. $\qquad\square$

## E EXPERIMENTAL DETAILS

### E.1 ANONYMOUS CODE

```
https://anonymous.4open.science/r/ICLR-2022-Anonymous-Demo-Code-B9FF/
README.md
```

## E.2 DLAE AND HYPERPARAMETER TUNING

This section presents the hyperparameter tuning process on the validation data over three (ML-20M, Netflix, MSD) datasets for the full-rank DLAE formula (case 6 in Table 1), which was introduced by Steck (2020) yet not investigated:

$$\min_{W} ||X - XW||_F^2 + ||\Lambda^{1/2} \cdot W||_F^2$$
$$\Lambda = \frac{p}{1-p} dMat(diag(X^T X))$$
$$\widehat{W} = (X^T X + \Lambda)^{-1} X^T X$$

In practical, $l_2$ regularization (hyperparameter $\lambda$) is also imposed:

$$\widehat{W} = (X^T X + \Lambda + \lambda)^{-1} X^T X$$

The Tables 5 to 7 show the results of $nDCG@100$ over three datasets respectively. And the optimal parameters are highlighted.

Table 5: ml-20m, DLAE full rank, parameter tuning on validation dataset by $nDCG@100$

|   |     | \multicolumn{6}{c}{$\lambda$} | | | | | |
|---|-----|---------|---------|-----------|---------|---------|---------|
|   |     | 800     | 900     | **1000**  | 1100    | 1200    | 1300    |
|   | 0.1 | 0.42024 | 0.42063 | 0.42073   | 0.42102 | 0.42131 | 0.4212  |
|   | 0.2 | 0.43132 | 0.43139 | 0.43154   | 0.4314  | 0.43147 | 0.43136 |
| p | **0.3** | 0.43203 | 0.43211 | **0.43214** | 0.43206 | 0.43203 | 0.43196 |
|   | 0.4 | 0.43001 | 0.43001 | 0.42995   | 0.42996 | 0.42984 | 0.42978 |
|   | 0.5 | 0.42754 | 0.42745 | 0.42729   | 0.42718 | 0.42715 | 0.42704 |

Table 6: netflix, DLAE full rank, parameter tuning on validation dataset by $nDCG@100$

|   |      | \multicolumn{7}{c}{$\lambda$} | | | | | | |
|---|------|---------|---------|---------|-----------|---------|---------|---------|
|   |      | 800     | 900     | 1000    | **1100**  | 1200    | 1300    | 1400    |
|   | 0.2  | 0.3904  | 0.3904  | 0.39027 | 0.39024   | 0.3902  | 0.3903  | 0.39018 |
|   | 0.25 | 0.39247 | 0.39252 | 0.39248 | 0.39249   | 0.3925  | 0.3925  | 0.39256 |
|   | 0.3  | 0.39359 | 0.39359 | 0.39366 | 0.39358   | 0.39362 | 0.39368 | 0.39369 |
| p | **0.35** | 0.39402 | 0.39403 | 0.394  | **0.39405** | 0.39399 | 0.39403 | 0.39397 |
|   | 0.4  | 0.39399 | 0.39393 | 0.39395 | 0.39393   | 0.39389 | 0.39388 | 0.39387 |
|   | 0.45 | 0.39346 | 0.3935  | 0.39343 | 0.39344   | 0.39338 | 0.3933  | 0.39329 |
|   | 0.5  | 0.39249 | 0.39241 | 0.39247 | 0.39241   | 0.39242 | 0.3923  | 0.39224 |

## E.3 MATRIX FACTORIZATION WITH DROPOUT AND HYPERPARAMETER TUNING

Cavazza et al. (2018) show that optimization with dropout (allowing rank $d$ to be optimized) is equivalent to solving a matrix approximation problem with nuclear norm:

$$\min_{P,Q,d} ||X - PQ||_F^2 + d\frac{1-p}{p} \cdot \sum_{k=1}^{d} ||P_k||_2^2 \cdot ||Q_k^T||_2^2$$
$$\min_{Y} ||X - Y||_F^2 + \frac{1-p}{p} ||Y||_*^2$$

(47)

Table 7: msd, DLAE full rank, parameter tuning on validation dataset by $nDCG@100$

|   |     | \multicolumn{6}{c}{$\lambda$} | | | | | |
|---|-----|---------|---------|-----------|---------|---------|---------|
|   |     | 10      | 20      | **30**    | 40      | 50      | 60      |
|   | 0.3 | 0.38514 | 0.38515 | 0.38517   | 0.38505 | 0.38492 | 0.38474 |
|   | **0.4** | 0.38596 | 0.38599 | **0.38602** | 0.386  | 0.38597 | 0.38592 |
| p | 0.5 | 0.38556 | 0.38555 | 0.38553   | 0.38557 | 0.38553 | 0.38549 |
|   | 0.6 | 0.38382 | 0.3838  | 0.38381   | 0.38374 | 0.38373 | 0.38366 |

and the solution is given by:

$$X \stackrel{\text{SVD}}{=} U\Sigma V^T$$
$$Y^* = P^* \cdot Q^*$$
$$= U \cdot S_\mu(\Sigma) \cdot V^T$$
$$S_\mu(\sigma) = \max(\sigma - \mu, 0)$$
$$\mu = \frac{1-p}{p + (1-p)\bar{d}} \sum_{i=1}^{\bar{d}} \sigma_i(X)$$

where $\bar{d}$ denotes the largest integer such that:

$$\sigma_{\bar{d}}(X) > \frac{1-p}{p + (1-p)\bar{d}} \sum_{i=1}^{\bar{d}} \sigma_i(X)$$

Hence, there is only one parameter $p$ to tuning. We present the tuning process on the validation set below, see Tables 8 to 10. Optimal parameter $p$ as well as induced rank $\bar{d}$ is highlighted.

Table 8: ml-20m, matrix factorization with dropout, hyper parameter tuning by $nDCG@100$ on validation dataset and its induced rank .

| p | 0.9 | 0.99 | 0.995 | **0.996** | 0.997 |
|---|---|---|---|---|---|
| induced rank $d$ | 10 | 200 | 385 | **467** | 602 |
| nDCG@100 | 0.29723 | 0.39369 | 0.40045 | **0.40046** | 0.39925 |

Table 9: netflix, matrix factorization with dropout, hyper parameter tuning by $nDCG@100$ on validation dataset and its induced rank .

| p | 0.9 | 0.99 | 0.996 | **0.997** | 0.998 |
|---|---|---|---|---|---|
| induced rank $d$ | 9 | 209 | 524 | **653** | 883 |
| nDCG@100 | 0.26026 | 0.35462 | 0.36453 | **0.36495** | 0.36406 |

Table 10: msd, matrix factorization with dropout, hyper parameter tuning by $nDCG@100$ on validation dataset and its induced rank .

| p | 0.99 | 0.999 | 0.9995 | **0.9999** | 0.99995 |
|---|---|---|---|---|---|
| induced rank $d$ | 249 | 2054 | 3783 | **11380** | 19308 |
| nDCG@100 | 0.18986 | 0.28532 | 0.307 | **0.32634** | 0.30995 |

## E.4 RESOURCES

Our code are mainly implemented in Numpy 1.19, Pytorch 1.7.1 on CUDA 11.0. Our experiments are performed on nodes with two sockets, each containing a 24-core Intel(R) Xeon(R) Platinum 8268 CPU @ 2.90GHz and 4 GeForce RTX 3090 24GB memory GPU.

