# OpenReview forum: "On the regularization landscape for the linear recommendation models"
_ICLR.cc/2022/Conference — ICLR 2022 Submitted_

### Official Review · Reviewer_7nvQ · 2021-10-23

**Correctness:** 3
**Technical Novelty And Significance:** 3
**Empirical Novelty And Significance:** 3
**Recommendation:** 6
**Confidence:** 4

**Main Review:**

Overall, I think the flow of the paper could be improved. Many things are stated/claimed in Section 2, and then the justified several pages later. Essentially, this means that the reader has to memorize many things until they get explained later. For instance, in section 2.1 on nuclear-norm regularization, various Frobenius-norm regularizations are given, and the connection to the nuclear-norm is provided 2 pages later in proposition 2. Even more confusing in section 2.1 is that A3 / Eq 2 appears to be different from A1, A2, and A4  in that L_2 norm regularization of ||W_1||^2 is used in Eq 2, i.e., without the data matrix X. In contrast, A1, A2, and A4 apply L_2 norm regularization essentially to ||X * W_1||^2, or if P= X * W_1 then ||X * W_1||^2. This is eventually justified 3 pages later in corollary 1. I wonder if one could simply swap the ordering of sections 2 and 3 to improve the flow. At the moment, I feel that the reader has to read the paper twice, as section 2.1 only makes sense after reading section 3.

Section 2:

The authors categorize the various linear models according to the applied regularization (nuclear-norm vs Frobenius). To this end, they also need weighted regularization, and they generalize recent results (which seems somewhat straight forward, though).

It is not clear to me why A3 / Eq 2 is called denoising linear autoencoder, given that denoising in linear models is equivalent to applying an L_2 norm regularization of the form ||W_1 * W2||^2 (i.e. product), and not ||W_1||^2+||W_2||^2 (i.e., sum) (dropping the L_2 parameters here for concise notation). A second version of a denoising linear autoencoder appears in A6, where the correct L_2 regularization induced by denoising is used. Maybe this can be described more clearly.

I wonder if  the claim that Eq 5 is equivalent to MF is actually trivial, as one can simply identify P:= XA and Q:=B to connect both formulations. The equations below Eq 11 in Jin's paper already show that MF and autoencoders are equivalent models, i.e., P*Q in MF can be equivalently rewritten as an autoencoder taking the form X*A*B--hence the only difference and the key difference is the L_2 norm regularization ||P||^2 = ||XA||^2 (in MF) vs ||A||^2 (in the autoencoder), i.e., X is included vs excluded in the L_2 norm regularization. Hence, it is obvious that one could use the regularization  ||XA||^2 in the autoencoder, which would obviously be equivalent to MF. Apart from that, it is not clear to me why the weighting is introduced, as it  does not seem to add anything to the conceptual insights gained (but I can believe that it will lead to better results in practice/experiments)

In A7 (EDLAE), the statement is incorrect that the constraint diag(W)=0 is applied to low-rank EDLAE models. Instead, the diagonal is subtracted during training, i.e., || X- X * {W-diagMat(diag(W)) } ||^2. In other words, while W is not required to have a 0 diagonal, the matrix {W-diagMat(diag(W)) } obviously has a zero diagonal. If W is low rank, then the diagonal is generally not 0, as the diagonal and off-diagonal elements are not independent of each other (due to the low-rank constraint). Only when W is full-rank, all entries in W become independent of each other, so that one can afford the freedom to constrain the diagonal to 0.

Section 3:

This section provides the connection between nuclear norm and Frobenius norm regularization. The key contribution seems to be that existing results on the nuclear norm are generalized to the *weighted* nuclear norm.

Section 4:

Below Eq 13, it says that diag(W)=0 in EDLAE. This is only true for the full-rank EDLAE model, while it is not true for low-rank EDLAE, which is considered here. See comment above.

This section provides 2 ways of approximating the full-rank EDLAE by a low-rank model, which somewhat seems to fall out of the blue sky, i.e., it seems unrelated to the previous parts of the paper--including the section 2.2 on Frobenius norm regularization applied to various models. While it is clearly outlined how the 2 methods work, it is not discussed *why* they can be expected to work / result in accurate approximations. The reason is that the diagonal is ignored / assumed away in both proposed methods, even though the diagonal is essential in the original EDLAE model. It would be great if the authors could provide some theoretical / intuitive justification why this approximation is valid/accurate.  The experiments show that this approach indeed works surprisingly well.

It is not clear to me what it means that the best of both  nuclear norm and Frobenius norm regularization are obtained by these 2 methods. In this generality, I think this is not true.  For instance, Jin et al. 2021 derived a closed form solution of A6, which uses Frobenious norm regularization and is a low-rank model. Hence, obtaining a closed-form solution is not a property of the nuclear-norm regularization, as suggested/implied by the authors. The authors propose 2 methods for a closed form approximation to EDLAE--which uses Frobenius norm regularization--it is hence not clear to me why there is any need to make a connection to the  nuclear norm regularization in this section. The only remaining indirect justification could be that nuclear-norm regularization induces a low-rank model, but conversely, I am not sure that a low-rank model induces the need for nuclear-norm regularization. Moreover, the full-rank EDLAE model is usually  more accurate than the low-rank EDLAE--from this perspective there may not be any need for  nuclear-norm regularization or the low-rank version of EDLAE.

Section 5:

The experiments show that the 2 proposed methods achieve high accuracy.

As a suggestion, Table 1 might be easier to read if it were explicitly indicated which models are full-rank, and which are low-rank (and which rank k).

Additional points:

- Given that this paper makes several contributions that are not all connected to each other, I wonder if it would make sense to split this paper up, as it would allow for more space to discuss the various contributions in more depth.

- The paper contains many typos and grammatical mistakes. Also note that 'non-descending' = 'ascending'.

- below Eq 8, are Q' and P' swapped? I.e., I would have expected P' = XA^*, not Q'XA^*.

- Lambda in A3 and A6 seem to be different matrices--this should be reflected in the notation.

++++++++++++++++++++++++++++++++++++++++++++++++++++++++++++++++++++++++++++++++++
Update based on authors' responses:
I very much appreciate the detailed responses and improvements in the paper. As I expected that many of the improvements can be made easily, and factored them into the score (which is the highest among all the reviews), I would like to maintain my current score.
As an aside, I am not really convinced by the arguments for dropping the zero diagonal constraint for the low-rank model, as it is unclear to me how tight the bounds are, and hence it is unclear if optimizing the bound  really yields similar results as optimizing the original objective (with the zero diagonal). Moreover, when dropping the zero-diagonal constraint, the bound seems to be a lower (rather than an upper) one, which means that a lower bound is minimized.

**Summary Of The Paper:**

This paper categorizes various linear models based on the applied regularization: nuclear norm vs Frobenius norm regularization. It is claimed that this categorization is key to understanding the prediction accuracies of the various linear model-classes. The paper also makes connections between nuclear norm regularization and Frobenius-norm regularization, and also discusses L_p norm regularization of the singular values. The paper also describes 2 methods that yield a closed-form solution of a low-rank model with Frobenius norm regularization: while it is not motivated/explained why these methods should work from a conceptual/theoretical perspective, the experiments show that they indeed work surprisingly well.


**Summary Of The Review:**


postive points:

- results on unweighted matrix norms are generalized to the *weighted* nuclear norm, connecting the latter to the weighted Frobenius norm.
- various models are categorized based on the used regularization (nuclear norm vs Frobenius norm), and it is shown that this explains the differences in prediction accuracy to a large degree.
- the proposed 2 methods / approximations work well in the experiments (but are not motivated/justified conceptually).

negative points:

- the 2 methods/approximations in section 4 are not motivated / justified theoretically (but they work well in the experiments).
- the writing and the flow of the paper could be improved considerably.

---

> ### Author Response · Authors · 2021-11-19
> **Response to Reviewer 7nvQ(part 1/3)**
>
> We sincerely appreciate the reviewer’s positive feedback , valuable comments and suggestions . Below, we provide our response to the reviewer's detailed concerns.
>
> **Q1. Overall, I think the flow of the paper could be improved. Many things are stated/claimed in Section 2, and then the justified several pages later. Essentially, this means that the reader has to memorize many things until they get explained later. For instance, in section 2.1 on nuclear-norm regularization, various Frobenius-norm regularizations are given, and the connection to the nuclear-norm is provided 2 pages later in proposition 2. Even more confusing in section 2.1 is that A3 / Eq 2 appears to be different from A1, A2, and A4 in that $L_2$ norm regularization of $||W_1||^2$ is used in Eq 2, i.e., without the data matrix X. In contrast, A1, A2, and A4 apply $L_2$ norm regularization essentially to $||X * W_1||^2,$ or if $P= X * W_1$ then $||X * W_1||^2$. This is eventually justified 3 pages later in corollary 1. I wonder if one could simply swap the ordering of sections 2 and 3 to improve the flow. At the moment, I feel that the reader has to read the paper twice, as section 2.1 only makes sense after reading section 3.**
>
> **A1.** Thanks to the valuable comments again, and we make some changes to the organization of our paper.
> Briefly, for section 2,  we summarize the background of previous work - nuclear-norm-based regularization models and  Frobenius-norm-based regularization models. Note, the unification of the existing models could provide the reader a overall landscape of our work. Consequently, we characterize these two different frameworks as well as their connections.
>   In section 3, we show the rigidity of the nuclear-norm-based regularizations and in
>   and section 4, we introduce two low-rank closed-form solution to the state-of-the-art (Frobenius-norm-based) linear recommendation model.
>
> **Q2. It is not clear to me why A3 / Eq 2 is called denoising linear autoencoder.**
>
> **A2.** We change the name of A3 to Weighted Linear Autoencoder (WLAE), where there exists a closed-form solution if the weight $\Lambda$ is non-descending.
>
> **Q3. I wonder if the claim that Eq 5 is equivalent to MF is actually trivial, as one can simply identify $P:= XA$ and $Q:=B$ to connect both formulations. The equations below Eq 11 in Jin's paper already show that MF and autoencoders are equivalent models, i.e., PQ in MF can be equivalently rewritten as an autoencoder taking the form $XA*B$--hence the only difference and the key difference is the $L_2$ norm regularization $||P||^2 = ||XA||^2$ (in MF) vs $||A||^2$ (in the autoencoder), i.e., X is included vs excluded in the $L_2$ norm regularization. Hence, it is obvious that one could use the regularization $||XA||^2$ in the autoencoder, which would obviously be equivalent to MF.**
>
> **A3.** Thanks for your insights on the question.
> In Jin's paper[1] (likely the equations 15?), they show for the optimal solution of LAE, it can be represented as a MF format with $P$ and $Q$. However, from MF regularization perspective, they didn't discuss or give any indication on how MF regularization can reach that optimal solution. They simply show there are user and item vector (matrices) which can produce the same results as LAE.  Also the general optimal LAE solution with Frobenius norm does not necessarily  mean it can be represented as a MF problem.
>
>  Now, in this problem, when we can set P=XA and Q=B, and let the syntax of Eq. 8 match with that of Eq. 5, it simply implies that the solution for Eq. 8 is a lower bound of that for Eq. 5. The main challenge is that we have to show the optimal $P$ in eq. (8) are spanned in the columns of X. We rigorously prove this result in Appendix D.5 (with an proof intuition in D.4).
>
> **Q4. Apart from that, it is not clear to me why the weighting is introduced, as it does not seem to add anything to the conceptual insights gained (but I can believe that it will lead to better results in practice/experiments)**
>
> **A4.** A key idea in a recent performance leader EDLAE is to utilize of *weighted/non-uniform regularizers* on the parameters' Frobenius norm. Applying dropouts[2] is shown to be equivalent to re-weighting the exponents in the regularizers, such as designing the weighted sum of regularizers based on other norms or tuning the exponent weights. We are also interested in determining what circumstance closed-form solutions still exist.

---

> > ### Author Response · Authors · 2021-11-19
> > **Response to Reviewer 7nvQ(part 2/3)**
> >
> > **Q5 In A7 (EDLAE), the statement is incorrect that the constraint diag(W)=0 is applied to low-rank EDLAE models. Instead, the diagonal is subtracted during training, i.e., $|| X- X * {W-diagMat(diag(W)) } ||^2$.**
> >
> > **A5.** To be more accurate, we match our expression of low-rank EDLAE to be same as [3]. In addition, As the rank $k$ is small, we subtract the diagonal of $W$ in objective; when $k$ is sufficient large (also suggested in [3], we can approximate the objective by explicitly enforcing the zero-diagonal constraint on the local rank matrix.
> >
> > **Q6. This section(3) provides the connection between nuclear norm and Frobenius norm regularization. The key contribution seems to be that existing results on the nuclear norm are generalized to the weighted nuclear norm.**
> >
> > **A6.** In the revision, section 3 highlights our results on  nuclear-norm-based regularizations. This section presents two results. *(i)* Eq. (5) (A4) is equivalent to a model with weighted nuclear-norm, where the weights are diagonal values of $\Lambda$ arranged in non-descending order. Because weighted nuclear norms are non-differentiable in general so Eq. (5)  compiles  non-differentiable objectives into differentiable ones, which are easier to optimize. Also, the auto-sorting property restricts Eq. (5)  from expressing an arbitrary weight sequence in $||W||$ $\omega$,$\*$, which limits its predictive power. *(ii)* We give a closed-form solution for models with regularizer $\|W\|^p_*$ for all $p \geq 1$. The solutions share the same structure so tuning $p$ will not improve a model's predictive power.
> >
> > **Q7.This section 4 provides 2 ways of approximating the full-rank EDLAE by a low-rank model, which somewhat seems to fall out of the blue sky, i.e., it seems unrelated to the previous parts of the paper--including the section 2.2 on Frobenius norm regularization applied to various models. While it is clearly outlined how the 2 methods work, it is not discussed why they can be expected to work / result in accurate approximations. The reason is that the diagonal is ignored / assumed away in both proposed methods, even though the diagonal is essential in the original EDLAE model. It would be great if the authors could provide some theoretical / intuitive justification why this approximation is valid/accurate. The experiments show that this approach indeed works surprisingly well.**
> >
> > **A7.** We believe these work are naturally connected to each other and we describe more about our motivation and intuition in the introduction section. For the proposed low-rank methods, We add more details in Appendix B that describe the analysis. We would like to claim that the proposed low-rank closed-form solution try to solve a bounding objective of the low-rank EDLAE problem. Here are we briefly explain a little bit:
> >
> > Recall that that low-rank EDLAE aims to solve
> >
> > \begin{equation*}
> >      \begin{split}
> >          W &= \arg\min_{rank(W)\le k }||X-X(W-dMat(diag(W))||^2_F+||\Lambda^{\frac{1}{2}}(W-dMat(diag(W))||^2_F\\
> >      \end{split}
> >  \end{equation*}
> >
> >  When the rank $k$ of the low-rank model is sufficiently large (for instance $k \approx 1000$), its objective function can be approximate as [3]:
> > \begin{equation}
> >      \begin{split}
> >          W &= \arg\min_{rank(W)\le k }||X-XW||^2_F+||\Lambda^{\frac{1}{2}}W||^2_F\\
> >          s.t.&\quad diag(W)=0
> >      \end{split}
> >  \end{equation}
> >
> > Note that for most of our experimental studies, the rank $k$ in the low-rank model is typically fairly large (over $1K$), and thus the above approximation is reasonable.
> >  Our goal is to mimic the strategy developed in [1]  to build a two-staged algorithm to produce a low-rank approximation. And using our proposed Lemma 3, we have
> >
> >
> > $$    \min_{rank(W)\leq k} ||\overline{Y}-\overline{X}W||_F^2\\         \quad s.t.\quad diag(W)=0$$
> >
> >
> > $$    =\min_{rank(W) \leq k}  ||\overline{Y}-\overline{X} W^*||_F^2 + ||\overline{X}W^*-\overline{X}W||_F^2,  \quad s.t.  \quad diag(W^*)=0 \quad diag(W)=0 $$
> >
> >  $$ \ge \min_{rank(W) \leq k}  ||\overline{Y}-\overline{X} W^*||_F^2 + ||\overline{X}W^*-\overline{X}W||_F^2,   \quad s.t.  \quad diag(W^*)=0 $$
> >
> >
> > The last inequality holds because one constraint $diag(W) = 0$ is removed so the objective will only improve. With the above analysis, we are ready to describe our algorithm.

---

> > > ### Author Response · Authors · 2021-11-19
> > > **Response to Reviewer 7nvQ(part 3/3)**
> > >
> > > **(Subproblem 1:)** full-rank closed-from solution:
> > >
> > >
> > > $$ W^*=\arg\min ||\overline{Y}-\overline{X} W^*||_F^2$$
> > >
> > > $$= \arg\min_{W }||X-XW^*||^2_F+||\Lambda^{\frac{1}{2}}W^*||^2_F, \quad s.t. \quad diag(W^*)=0$$
> > >
> > >
> > >  The closed-form solution is:
> > >
> > >  \begin{equation*}
> > > \begin{split}
> > >         C&=(X^TX+\Lambda)^{-1}\\
> > >       W^*&= I - C\cdot dMat(1\oslash diag(C))
> > > \end{split}
> > > \end{equation*}
> > >
> > > **(Subproblem 2:)** low-rank matrix approximation:
> > >
> > > $$ \widehat{W} = \arg\min_{rank(W) \leq k} ||\overline{X}W^*-\overline{X}W||_F^2 $$
> > >
> > >  $$ = \arg \min_{rank(W) \leq k} ||X W^* - XW||_F^2+ ||\Lambda^{\frac{1}{2}} (W^*- W)||_F^2$$
> > >
> > >
> > > Note, in the subproblem 2, we relax the zero-diagonal constraint. Given this, the closed-from solution (LR-EDLAE-1) is given by:
> > > \begin{equation*}
> > > \boxed{
> > > \widehat{W}= W^*(Q_k Q_k^T)
> > > }
> > > \end{equation*}
> > > where $Q_k$ comes from:     $\overline{X}W^*\stackrel{\text{SVD}}{=}P \Sigma Q^T$.
> > >
> > >
> > > **Intuition on the efficacy of the proposed approximation.**
> > > We believe that because the full-rank $W^*$ performs well for recommendation accuracy and better than the low-rank solutions [1][2],  when $W$'s estimation is closer to $W^*$ with respect to the final recovery of $Y$, $XW \approx XW^*$, it has a comparable performance against $XW^*$.
> > > In other words, it may not be essential for the diagonal of low-rank approximate matrix $\widehat{W}$ to be zero.
> > >
> > > Furthermore, if we consider the inequality
> > >
> > >
> > > $$\min_{rank(W) \leq k}  ||\overline{X}W^*-\overline{X}W||_F^2   $$
> > >
> > > $$\leq \min_{rank(W) \leq k}  ||\overline{X}||_F^2||W^*-W||_F^2$$
> > >
> > > or a tighter one
> > >
> > >
> > > $$ \min_{rank(W) \leq k}  ||\overline{X}W^*-\overline{X}W||_F^2$$
> > >
> > > $$\leq \min_{rank(W) \leq k}  ||\overline{X}||_2^2||W^*-W||_F^2$$
> > >
> > > Both suggest if we simply perform SVD on $W^*$, we can obtain an upper bound error between $\overline{X}W^*$ and $\overline{X}W$.
> > >
> > >
> > > Simply, the other closed-form solution $\widehat{W}$ (LR-EDLAE-2) is:
> > >
> > > $$W^*\stackrel{\text{SVD}}{=}U\Sigma V^T$$
> > >
> > >  $$\widehat{W}=U_k\Sigma_k V^T_k$$
> > >
> > >
> > >
> > > **Q8. It is not clear to me what it means that the best of both nuclear norm and Frobenius norm regularization are obtained by these 2 methods.**
> > >
> > > **A8.** Sorry for the confusion, we have removed this discussion on  best of both nuclear-norm and Frobenius-norm regularization. Thanks for pointing this out; we agree with you the argument is a bit too convoluted.
> > >
> > > **Q9. As a suggestion, Table 1 (now table 2) might be easier to read if it were explicitly indicated which models are full-rank, and which are low-rank (and which rank k).**
> > >
> > > **A9.**  As suggested, we mark the full and low rank for (Frobenius-norm and nuclear-norm) linear models. Due to the space limitation, its hard to mark the low rank $k$ in the table while we explicitly point out in appendix (code).
> > >
> > > **Q10. Given that this paper makes several contributions that are not all connected to each other, I wonder if it would make sense to split this paper up, as it would allow for more space to discuss the various contributions in more depth.**
> > >
> > > **A10.** We believe these work are naturally connected to each other. Please see the new introduction on how these problems are connected together and motivate our study.
> > >
> > > **Q11.** The paper contains many typos and grammatical mistakes. Also note that 'non-descending' = 'ascending'.
> > > below Eq 8, are Q' and P' swapped? I.e., I would have expected $P' = XA^*$, not $Q'=XA^*$.
> > > Lambda in A3 and A6 seem to be different matrices--this should be reflected in the notation.**
> > >
> > > **A11.** We appreciate your kind notes and revise our expressions accordingly.
> > > - To be more rigorous, we use “non-descending” instead of “ascending”, since there could exist some identical consecutive terms in the ordered sequence.
> > > - We notice that P’ and Q’ are typos and correct them.
> > > - In general, $\Lambda$ refers to the (hyperparameter) diagonal matrix as a coefficient of the regularizer. We maintain this convention to address the motivation that can we apply the weighted regularization from Frobenius-norm-based model (EDLAE) to other nuclear-norm-based models . Specifically, if not explicitly stated, $\Lambda$ denotes the (hyperparameter) diagonal matrix. We thank the reviewer’s suggestions, and we highlight this claim in the caption of the summary table (table 1).
> > >
> > >
> > > **References:**
> > >
> > >
> > > [1] Ruoming Jin, et al. "Towards a better understandingof linear recommendation models." KDD 2021.
> > >
> > > [2] Cavazza, Jacopo, et al. "Dropout as a low-rank regularizer for matrix factorization." International Conference on Artificial Intelligence and Statistics. PMLR, 2018.
> > >
> > > [3] Steck, Harald. "Autoencoders that don't overfit towards the Identity." NeurIPS. 2020.

---

### Official Review · Reviewer_Uxog · 2021-11-02

**Correctness:** 3
**Technical Novelty And Significance:** 2
**Empirical Novelty And Significance:** 2
**Recommendation:** 5
**Confidence:** 3

**Main Review:**

The paper is well structured and easy to follow. The analyses from the perspective of regularization terms are novel. The results in Table 3 are very interesting. Although it is unknown why not ordering the singular values will lead to significant performance drop, the results can help confirm another limitation of matrix factorization-based methods.

My main concerns about this paper are as follows:
1)	This paper may help researchers to understand linear recommendation methods from a new perspective, but the theoretical contributions are limited. In addition, the proposed methods are without statistically significant improvements compared to baseline methods.
2)	The authors claimed that the proposed method can have closed form solutions. But in both proposed methods, SVD is required to obtain the so-called closed form solution. I am not sure if this kind of solutions are “closed form” or not.
3)	There is no efficiency analysis about the proposed method. Although the proposed methods are low-rank not full-rank, the required ranks are actually very high, e.g., 2000-12000 in Figure 1. This is rarely seen in practice and might cause storage or computation issue in large datasets.
4)	It will be interesting to see more insights about why ordering of weights on matrix factorization and other regularizations are so important in recommendation models.

**Summary Of The Paper:**

This paper analyzed the linear models for recommendation from their regularization terms, more specifically, via nuclear norm and Frobenius norm. Based on the analyzes, this paper proposed two alternatives for linear recommendation models with closed form solutions. Experiments on three large scale datasets showed that the proposed alternatives are comparable to the state-of-the-art baseline methods.

**Summary Of The Review:**

Overall, I feel that this paper makes incremental contribution by connecting existing recommendation methods both theoretically and empirically.

---

> ### Author Response · Authors · 2021-11-19
> **Response to Reviewer Uxog  (part 1/2)**
>
> We thank the reviewer for the feedback and valuable comments. Below we respond to specific questions.
>
> **Q1. This paper may help researchers to understand linear recommendation methods from a new perspective, but the theoretical contributions are limited.**
> **Overall, I feel that this paper makes incremental contribution by connecting existing recommendation methods both theoretically and empirically.**
>
> **A1.** Thanks for the comments. Here, we would like to summarize and emphasize our contributions:
>
> In this paper, we go much beyond the study in the latest research work [3],  which focuses on only two basic models, and analyze a large number of recent performance leaders of (linear) recommendation algorithms. We aim to provide an in-depth understanding of various regularized objective functions and towards unifying them through their closed-form solutions. In return, the closed-forms serve as the barebones engine to help reveal what drives the performance improvement for the recent recommendation algorithms.
>
> We examine and resolve three open and closely related questions:
>
>  **(i) Characterization of models.**
>
>  **(ii) Weighted generalization of regularizers.**
>
> **(iii) Low-rank closed-form solutions for EDLAE.**
>
> We believe these problems are significant and solving them are non-trivial.
> Our investigation also leads to the following discovery and resolves the above questions:
>
> **Regularizer dichotomy (section 2):**
> We found that all the leading (linear) recommendation models  can be categorized into those that implement possibly weighted *nuclear-norm* regularizers, and into those that implement *Frobenius-norm* regularizers. Specifically,
> we characterize the Variational Linear Autoencoders (VLAE) as a form of weighted nuclear-norm regularization problem, in which the weights possess a specific combinatorial structure.
> In addition, we observe that it is not matrix factorization or LAE that determines the shrinkage structure (as [3] suggested), but instead it is the form of regularization.
> Thus, this paper provides a more complete and accurate characterization on how a linear recommendation model performs under different regularizations.
>
> **Rigidity of nuclear-norm regularizers (section 3):**
>  With the dichotomy result, we next aim to understand whether the weighted sum idea for Frobenius norm regularizers can be generalized to nuclear-norm regularizers, and whether tuning exponent weights can be beneficial. First, while VLAE is equivalent to nuclear-norm regularization and is easier to optimize (weighted nuclear norms are non-smooth but VLAE's objectives are smooth), its closed-form solution possess a (surprisingly) rigid structure, i.e., the weighted regularization will lead to an auto-sorting singular value reduction and the larger single values tend to receive smaller reduction (*non-ascending reduction*). Second, it has been shown that the solution structure for a model with a squared nuclear-norm regularizer $||W||^2_\*$ (i.e., dropout's equivalence) is strikingly similar to that for using $||W||$\* nuclear-norm regularizer. We generalize the result to show that the solution structures for $||W||^p_*$ are highly similar for all $p \geq 1$ (Regularization invariant/rigidity with respect to $p$). But when $p = 1, 2$, the solution and hyperparameters possess favorable properties so hyperparameter search becomes easier.
> This also partially explains why only $p = 1, 2$ have been extensively considered.
> These rigidity properties severely limit the search space and explains why models that use only nuclear-norm regularizers share the same performance ceiling even when hyperparameters are extensively searched.
>
>
> **Closed-form solution for low rank EDLAE (section 4):** The (weighted) Frobenius-norm regularizers $||\Lambda W||^2_F$ ($\Lambda$ is the hyperparameter diagonal matrix) are implemented in EASE and  EDLAE [4]. These models produce closed-form full-rank estimators; and if the zero diagonal constraint on $W$ is enforced,  their singular vectors will no longer coincide with those of the data, and can deliver (slightly) better performance. However, no closed-form solution for the low-rank estimator is known and the current approaches rely on ADMM or Alternating Least Square.
> In this paper, we propose two new low-rank, closed-form estimators that deliver comparable results to the full-rank models (EASE and full-rank EDLAE) as well as the ADMM-based solutions.
> The new closed-form solutions abstract out all the computation nuance (e.g., the need to tune ADMM and deal with local optimal) and can help us compare different regularizations analytically.

---

> > ### Author Response · Authors · 2021-11-19
> > **Response to Reviewer Uxog (part 2/2)**
> >
> > **Q2. In addition, the proposed methods are without statistically significant improvements compared to baseline methods.**
> >
> > **A2.** It's worth noting that the core of the paper is to investigate the low-rank space and help build the universe framework instead of proposing SOTA algorithms with better performance. Besides, without parameter tuning (unlike existing ADMM low-rank methods), the proposed closed-form solution could exhibit convenience empirically and help understanding the essence of the problem. Besides the scalability benefit, we believe the meaning of low-rank closed-form solutions goes much beyond the performance. These closed-form formulas would also allow us to compare the different methods theoretically under the same page.
> >   Finally, we would like to address again on the advantage of having  closed-form solutions: they are easy to implement and use, and can be quickly tested using any standard matrix computation platform without the specialized recommendation library. They are more robust to error and bug, and can serve as the competitive baseline for recommendation evaluation tasks.
> >
> > **Q3. The authors claimed that the proposed method can have closed form solutions. But in both proposed methods, SVD is required to obtain the so-called closed form solution. I am not sure if this kind of solutions are “closed form” or not.**
> >
> > **A3.** We note that the methods leveraged SVD could be treated as the closed-from, regularized PCA [1], MF dropout [2], etc.
> >
> > **Q4. There is no efficiency analysis about the proposed method. Although the proposed methods are low-rank not full-rank, the required ranks are actually very high, e.g., 2000-12000 in Figure 1. This is rarely seen in practice and might cause storage or computation issue in large datasets.**
> >
> > **A4.** We add the time complexity analysis in Appdendix C. Specifically, running time for full-rank Frobenius-norm-based models (EASE, DLAE, EDLAE) is $O(n^{2.373})$; for EDLAE-ADMM method is $O(tn^{2.373}$) ($t$ is the iteration of ADMM optimization) and for our newly proposed low-rank methods is $O(n^{2.373}+kn^2)$ ($k$ is the low rank). In practice, solving low-rank SVD and computing matrix multiplication require similar time especially after carefully using HPC techniques. So our proposed algorithms are no slower or faster than existing ones that do not use gradient algorithms. The running time for gradient-based algorithms is usually worse than that in theory because determination of hyper-parameters such as learning rate is usually a guesswork.
> >
> > Finally, in many real world situations, the number of items can easily reach hundreds of thousands to millions, local rank solutions are essential to handle such scalability with respect to the number of large items.
> >
> > **Q5.** It will be interesting to see more insights about why ordering of weights on matrix factorization and other regularizations are so important in recommendation models.
> >
> > **A5.** A key idea in a recent performance leader EDLAE is to utilize of {\em weighted/non-uniform regularizers} on the parameters' Frobenius norm. Applying dropouts is shown to be equivalent to re-weighting the exponents in the regularizers, such as designing the weighted sum of regularizers based on other norms or tuning the exponent weights. We are also interested in determining what circumstance closed-form solutions still exist.
> >
> > **References:**
> >
> > [1] Zheng, Shuai, Chris Ding, and Feiping Nie. "Regularized singular value decomposition and application to recommender system." arXiv preprint arXiv:1804.05090 (2018)
> >
> > [2] Cavazza, Jacopo, et al. "Dropout as a low-rank regularizer for matrix factorization." International Conference on Artificial Intelligence and Statistics. PMLR, 2018.
> >
> > [3] Ruoming Jin, et al. "Towards a better understandingof linear recommendation models." KDD 2021.
> >
> > [4] Steck, Harald. "Autoencoders that don't overfit towards the Identity." NeurIPS. 2020.

---

### Official Review · Reviewer_bVYo · 2021-11-02

**Correctness:** 3
**Technical Novelty And Significance:** 3
**Empirical Novelty And Significance:** 2
**Recommendation:** 5
**Confidence:** 4

**Main Review:**

1. My main concern about this work is that the performance of the proposed methods is not better than SOTA ones like DLAE and EDLAE. Also, the authors did not analyze or compare the computational complexity. I do not see the value of proposing a method with similar performance and unclear complexity. Specifically, the two proposed methods both require SVD approximation, which has a cubic complexity and is quite expensive. It is nontrivial to analyze the complexity of the proposed two methods and other algorithms.

2. The authors claim that ‘while the syntax of this problem resembles matrix completion (MC), recommendation systems and MC have different evaluation criteria so MC’s results are not directly applicable here.’ It is better and helpful to make it clear what the evaluation criteria difference is if the authors want to discuss the relations between recommender and MC. Besides, if two problems do not share the same evaluation criteria but have the same or similar objective function to be optimized, they are still quite correlated.

3. Section 3 has the same name as Section 2.1, which is confusing. The overall structure should be reorganized.

4. In Section Matrix Factorization via dropouts,  the relation of mu and X and lambda should be better explained.

Minor:
1. choise—> choice
2. Please unify ‘closed form’ and ‘’close-form’
3. which make the —> which makes the
Please polish the English writing and correct all grammar errors.

**Summary Of The Paper:**

The authors studied some linear recommendation algorithms and their relations and also proposed two new methods to benefit from both
nuclear-norm and Frobenius-norm regularizations. The two new methods have similar performance as compared methods.

**Summary Of The Review:**

Overall, the authors have done extensive works to analyze the SOTA linear recommendation models but some critical analysis like the computational complexity is missing. Also, the proposed methods do not perform better than SOTA ones.

---

> ### Author Response · Authors · 2021-11-19
> **Response to Reviewer bVYo (part 1/2)**
>
> We sincerely thank the reviewer for your comments and suggestions, please check our answers below.
>
> **Q1.My main concern about this work is that the performance of the proposed methods is not better than SOTA ones like DLAE and EDLAE.**
>
>   **A1.** The goal of the paper is to help reveal the "barebone" of the recommendation algorithmic engines and use that to help understand and compare different models.
>    Although its performance may not be better than its full-rank counterparts (EDLAE and DLAE), low-rank solutions are easier to interpret, use less storage, and can be more scalable with respect to the number of items. In addition, a closed-form solution disentangles key performance drivers from nuances (e.g., need to tune learning rate or deal with local optimal), and can help reveal the key driven factors comparing with other closed-form solutions.
>
>    Finally, we would like to point out some other advantages of the closed-form solutions: they are easy to implement and use, and can be quickly tested using any standard matrix computation platform without the specialized recommendation library. They are more robust to error and bug, and can serve as the competitive baseline for recommendation evaluation tasks.
>
> **Q2. Also, the authors did not analyze or compare the computational complexity. I do not see the value of proposing a method with similar performance and unclear complexity. Specifically, the two proposed methods both require SVD approximation, which has a cubic complexity and is quite expensive. It is nontrivial to analyze the complexity of the proposed two methods and other algorithms.**
>
> **A2.** We add the time complexity analysis in Appdendix C. Specifically, running time for full-rank Frobenius-norm-based models (EASE, DLAE, EDLAE) is $O(n^{2.373})$; for EDLAE-ADMM method is $O(tn^{2.373}$) ($t$ is the iteration of ADMM optimization) and for our newly proposed low-rank methods is $O(n^{2.373}+kn^2)$ ($k$ is the low rank). In practice, solving low-rank SVD and computing matrix multiplication require similar time especially after carefully using HPC techniques. So our proposed algorithms are no slower or faster than existing ones that do not use gradient algorithms. The running time for gradient-based algorithms is usually worse than that in theory because determination of hyper-parameters such as learning rate is usually a guesswork.
>
>  In the research area of recommender system, there is lacking of theoretical foundations and further suffering from non-unified or weak baselines. We would like to emphasize again that the proposed low-rank especially closed-form methods allow us to establish easy to implement baselines, unify and compare the different models theoretically and provide new perspective, which goes much beyond the performance and speed.
>
> **Q3. The authors claim that ‘while the syntax of this problem resembles matrix completion (MC), recommendation systems and MC have different evaluation criteria so MC’s results are not directly applicable here.’ It is better and helpful to make it clear what the evaluation criteria difference is if the authors want to discuss the relations between recommender and MC.  Besides, if two problems do not share the same evaluation criteria but have the same or similar objective function to be optimized, they are still quite correlated.**
>
>  **A3.** The problem is closely related to matrix completion (MC) because $X_{i,j} = 0$ can be viewed as "missing a data point" but MC's evaluation metrics is mean-squared error and is different from ours [1]. The connection between two problems results in models with similar objectives [2]. A technique developed for one problem often finds its counterpart for the other. Nevertheless, because of the difference in evaluation, efficacy of an algorithm for one problem does not imply any performance guarantee for the other. Thus, the non-overlapping component between two problems remains substantial. We also remark that our structural results on weighted nuclear-norm is new and applicable to MC.
>
> **Q4. Section 3 has the same name as Section 2.1, which is confusing. The overall structure should be reorganized.**
>
> **A4.** Thanks again for valuable comments. We reorganize the paper and claim our contributions to emphasize our work and navigate the paper flow at the end of introduction (Section 1) to exhibit a clear landscape. Specifically, we describe the background and regularization dichotomy (Frobenius-norm-based and nuclear-norm-based) in Section 2. We change the title of section 3 to "Rigidity of nuclear-norm-based regularization" to reveal the our study object and address our contribution.

---

> > ### Author Response · Authors · 2021-11-19
> > **Response to Reviewer bVYo (part 2/2)**
> >
> > **Q5. In Section Matrix Factorization via dropouts, the relation of mu and X and lambda should be better explained.**
> >
> >  **A5.** Due to the space limitation, we are unable to put the details of the equation/relation in the main context, we add the notes to refer to the corresponding appendix. We briefly explain a little bit here: The optimization with dropout (allowing rank optimizing) is equivalent to solving a matrix approximation problem with nuclear norm [3]:
> >
> > $$\min_{P, Q, d}{||X-PQ||^2_F+d\frac{1-p}{p}\cdot\sum\limits_{k=1}^d||P_k||^2_2 \cdot ||Q^T_k||^2_2}$$
> >
> > $$\min_{Y}{||X-Y||^2_F}+\frac{1-p}{p}||Y||^2_*$$
> >
> > Note in the main context A2, we simplify the parameter $\frac{1-p}{p}$ as $\lambda$. And the solution is given by [3]:
> >
> >  $$X \stackrel{\text{SVD}}{=} U\Sigma V^T$$
> >
> > $$Y^* = P^* \cdot Q^*=U \cdot S_\mu(\Sigma) \cdot V^T$$
> >
> > $$S_\mu(\sigma)=\max(\sigma - \mu, 0)$$
> >
> > $$\mu =\frac{1-p}{p+(1-p)\bar{d}}\sum\limits_{i=1}^{\bar{d}}{\sigma_i(X)}$$
> >
> > where $\bar{d}$ denotes the largest integer such that:
> > \begin{equation*}
> >     \sigma_{\bar{d}}(X)>\frac{1-p}{p+(1-p)\bar{d}}\sum\limits_{i=1}^{\bar{d}}{\sigma_i(X)}
> > \end{equation*}
> >
> > References:
> >
> > [1] Candès, Emmanuel J., and Terence Tao. "The power of convex relaxation: Near-optimal matrix completion." IEEE Transactions on Information Theory 56.5 (2010): 2053-2080.
> >
> > [2] Shuai Zheng and Chris Ding and Feiping Nie. Regularized Singular Value Decomposition and Application to Recommender System.
> >
> > [3] Cavazza, Jacopo, et al. "Dropout as a low-rank regularizer for matrix factorization." International Conference on Artificial Intelligence and Statistics. PMLR, 2018.

---

### Official Review · Reviewer_nwQT · 2021-11-02

**Correctness:** 4
**Technical Novelty And Significance:** 3
**Empirical Novelty And Significance:** 3
**Recommendation:** 5
**Confidence:** 3

**Main Review:**

Note to ICLR 2022 (not the authors): next time, it would be nice to include row number in the submission template to ease the reviewing process!

== STRENGTHS ==

The paper is rather well written with several emphases on the different contributions of the authors. There are solid theoretical foundations accompanied by empirical evidences on three real datasets.
The experiments include 1) recommendation performance (NDCG) of the several considered approaches, 2) influence of low rank value on NDCG and 3) impact of weight ordering on recommendation performance.
Moreover, the authors plan to share the code which is highly appreciable in order to provide reproducible results.

== WEAKNESSES (OR SUGGESTIONS) ==

There is a certain amount of minor typos (see some below) but also a lack of some term definitions (because of late copy-paste in Appendix to observe the page limitation rule probably).
Table 3 is actually very useful for the understanding, I would not put it in Appendix.

Some things which could be clarified:
- Reference to deep learning methods is done for state-of-the-approaches. In the paper, it seems it is more about shallow networks (max 2 layers) that are discussed.
- p.3, A3, Eq. 2: W1 and W2 are not defined. I guess they denote the Encoder and the Decoder network.
- p.3, A4, eq.3: W and V not defined, same as above
- Eq. 4: N is not defined
- p.4, A6: M not defined.
- Eq. 13, p.7: operation dMat not defined (too late to put it in p.2 of supplementary material), same in eq. for W* (which should be indexed)

Also to help the reader, it would be nice to start Sections 2 and 3 with a summary of what the section plans to achieve/demonstrate. In particular, this is quite difficult in Section 3 to follow the objective.
For instance, a sentence such as "Prop. 2 also leads to the following Corollary." does not help to understand the implication of such result.

Unless I missed it, differences of implication between LR-EDLAE-1 and LR-EDLAE-2 (the proposed methods) are not enough carefully detailed.

In Table 1, it would be clearer for the reader to identify the best methods for instance, 1) by putting in bold the best obtained results and 2) by underlying the 2nd best result.
Table 1 refers to Table 3 for more details: I think this is a mistake because it does not let the paper "self-contained".

Am I missing something or Mult-VAE/DAE are used as baselines during experiments but are not discussed before (contrarily to the other approaches)? Why?

Some typos :
- in abstract : « (surprisnig) »
- abstract: missing "-" for low-rank and closed form
- p.1 in introduction: "the linear autoencoders [...] which encompasses"
- missing upper case : p.2, 2nd paragraph « . we generalize the »
- p.2 missing "-", "These models produce closed form full-rank estimators"; "However, no closed form solutions"; "ADMM based solutions"; "the full rank W"
- end of p.2 : « The weighted unclear norm »
- beginning of p. 3 : missing lower case « Therefore, Nuclear-norm regularizers »
- inconsistency in convention naming for equations : eq. Or Eq. / eq (1) or eq. 1 ; 5 times in p. 3, same for Table or table, Proposition or Prop.
- p.3, A2 : « This approach useS »
- p.3, A4 : « probabolistic »
- p.4, first row: choose between "hyperparameter" (p.3 after Lemma 1) and "hyper-parameter" (p. 4)
- p.4, (ii): "choices of p [...] produces" ; "all entries in X is"
- p.4, (ii): inconsistency in naming convention: nuclear-norm-based and nuclear-norm based in the same sentence, sometimes it is nuclear norm (check everywhere in the paper)
- Before eq.7: "Its a closed-form solution is."
- choose between "Frobenius-norm regularizer" p.4,5 and "Frobenius norm regularizers" p.5
- before Sec.3, p.5: "two low-rank Frobenius-norm-based modelS"
- p.8 in Table 1: "Frobinius"
- eq.8, missing "|" in 3rd norm, extra space to remoce: "equivalently ,"
- missing "-" for closed form solutions after Proposition  2 p.5, in Section 4 p.7, in Section 5 p.7, in Q1 p.8, Q.2 in p.9 twice, Q.3, in conclusion ; check for low-rank everywhere also
- p.5 choose between "rearrangement" and "re-arrangement"
- Corollary 1, p.6: Missing Eq. before (9)
- Section 4, p.7: missing "-" for state-of-the-art
- Section 4, p.7: after ADMM: \cite instead of \citep, same after closed-form and EDLAE
- Table 1: refer TO
- p.8 in Q1: (eq. (11))
- p8, Q1: "one of the most popular implicit matrix factorization algorithmS"
- Q2: "most of them reaches"
- Q2, missing "-" in EDLAE based approaches
- Q2, p.9: ADMM method performS slightly better ; none-the-less
- p.9: "all the models add either a nuclear-norm-based [...] or a Frobenius-norm based regularizer." ; check also the abstract
- p.9 "The Frobenius-norm models are more express"
- Supplementary: section A autoencoder vs auto-encoder
-Supp, section A: \cite instead of \citep for 2nd paragraph






**Summary Of The Paper:**

This paper unifies under one theoretical framework the different state-of-the-art regularization approaches for linear collaborative-filtering recommendation models which leverage the user-item interaction data matrix.

The authors classify these algorithms among two families: 1) the nuclear-norm-based regularizers and 2) the Frobenius-norm-based regularizers. Both have their advantages and drawbacks.
While nuclear-norm-based regularizer's solutions are low-rank (and therefore more scalable) and have a closed form, their performance is limited.
On the other hand, Frobenius-norm-based regularizer's solutions are full-rank or difficult to train.
Hence, focusing on generalizing Frobenius-norm-based regularizers, the authors bridge the gap between the two families by providing two new low-rank, closed-forms solutions - LR-EDLAE-1 and LR-EDLAE-2 - peforming similarly to full-rank EASE and EDLAE state-of-the-art models.

To that purpose, the authors start from the analysis work from Jin et al. (2021) about the relationship between Matrix Factorization approaches and linear autoencoders-based ones.
Besides EASE and EDLAE algorithms, they also analyze VLAE, DLAE and LRR. They show that VLAE solves a weighted nuclear-norm regularixation problem.
They also generalize the result on the equivalence between dropout technique and the adding of a squared nuclear-norm regularizer to show that solution structures for ||W||^p_* are similar for all p>= 1.

**Summary Of The Review:**

This paper represents an extensive work on both theoretical and experimental sides.
However, the number of typos and missing notation definitions show that the manuscript has not been re-read enough to check that the fact of having moved parts from the general paper to appendix does not hurt actually the linear reading.
I advise the authors to do another reading pass to fix the typos and problems of definitions listed above.
I vote for a weak reject.

======  AFTER REBUTTAL ======
I read carefully the other reviews and author's responses. First of all, thank you very much to the authors for their hard work during the rebuttal period and the significative changes brought to the structure of the paper which is improving its quality a lot.
However, I tend to share the concerns of some reviewers regarding the complexity of the algorithm which is actually rather similar to the state-of-the-art baselines. Hence if the designed methods bring neither substantive improvements in terms of accuracy nor in terms of complexity, the proposed theoretical framework does not seem finally enough for an acceptance. I keep my grade "marginally below the acceptance threshold".

---

> ### Author Response · Authors · 2021-11-19
> **Response to Reviewer nwQT (part 1/2)**
>
> We sincerely thank the reviewer for your valuable comments and suggestions, please check our answers below.
>
>  **Q1. There is a certain amount of minor typos (see some below) but also a lack of some term definitions (because of late copy-paste in Appendix to observe the page limitation rule probably). Table 3 is actually very useful for the understanding, I would not put it in Appendix.**
>
> **A1.** We appreciate for the reviewer's constructive suggestions. We make several modifications to help understanding, specifically:
> - We fix the typos accordingly and unify multiple notations (e.g. "closed-form", "nuclear-norm-based", etc).
> - We add the explanations of notations when they appeared at first time.
> - We move the table 3 (now table 1 in the revised version) from appendix to Section 2 to overview the landscape of linear recommendation models with corresponding regularizations and (closed-form) solutions.
> - We summarize our contributions and present the paper structure at the end of introduction (Section 1) to help readers better navigate the paper.
> - As suggested, we start each section with a summary of what the section plans to demonstrate.
>
> **Q2. Some things which could be clarified: Reference to deep learning methods is done for state-of-the-approaches. In the paper, it seems it is more about shallow networks (max 2 layers) that are discussed.**
>
> **A2.** VAE-based recommendation models can have multiple layers, such as MultiVAE. Other deep learning models have shown be worse than linear ones [1]. Therefore, we primarily compare our models with VAE-based deep learning approaches.
>
> **Q3. In particular, this is quite difficult in Section 3 to follow the objective. For instance, a sentence such as "Prop. 2 also leads to the following Corollary." does not help to understand the implication of such result.**
>
> **A3.** After the revision, we highlight the major results of section 3, i.e., rigidity of the nuclear-norm-based regularization. Specifically, section 3 presents two results. *(i)* Eq. (5) (A4) is equivalent to a model with weighted nuclear-norm, where the weights are diagonal values of $\Lambda$ arranged in non-descending order. Because weighted nuclear norms are non-differentiable in general so Eq. (5) compiles  non-differentiable objectives into differentiable ones, which are easier to optimize. Also, the auto-sorting property restricts Eq. (5) from expressing an arbitrary weight sequence in  $||W||$ $\omega, \*$ , which limits its predictive power. *(ii)* We give a closed-form solution for models with regularizer $||W||^p_*$ for all $p \geq 1$. The solutions share the same structure so tuning $p$ will not improve a model's predictive power.
>
> We also clean up the claims and address the implications of the Proposition 2:
> *(i)* Both A3 and A4 effectively add a nuclear-norm regularizer. *(ii)* Diagonals of $\Lambda$ do not need to be sorted in ascending order as stated in [2] because any permutation of the diagonals will be equivalent to $OPT2$. This also limits the search space and affects a model's prediction power. *(iii)* Proposition 2 ``compiles'' a non-differentiable objective ($OPT2$) into an equivalent differentiable one ($OPT1$), which is easier to optimize. In addition, $f(\cdot)$ in A3 \& A4 is  the reconstruction error, in which case closed-form solutions exist.
>
>
> **Q4. Unless I missed it, differences of implication between LR-EDLAE-1 and LR-EDLAE-2 (the proposed methods) are not enough carefully detailed.**
>
> **A4.** For the proposed low-rank methods, We add more details in Appendix B that describe the analysis. We would like to claim that the proposed low-rank closed-form solution try to solve a bounding objective of the low-rank EDLAE problem. Here are we briefly explain a little bit:
>
> Recall that that low-rank EDLAE aims to solve
>
> \begin{equation*}
>      \begin{split}
>          W &= \arg\min_{rank(W)\le k }||X-X(W-dMat(diag(W))||^2_F+||\Lambda^{\frac{1}{2}}(W-dMat(diag(W))||^2_F\\
>      \end{split}
>  \end{equation*}
>
>  When the rank $k$ of the low-rank model is sufficiently large (for instance $k \approx 1000$), its objective function can be approximate as [3]:
> \begin{equation*}
>      \begin{split}
>          W &= \arg\min_{rank(W)\le k }||X-XW||^2_F+||\Lambda^{\frac{1}{2}}W||^2_F\\
>          s.t.&\quad diag(W)=0
>      \end{split}
>  \end{equation*}
>
> Note that for most of our experimental studies, the rank $k$ in the low-rank model is typically fairly large (over $1K$), and thus the above approximation is reasonable.

---

> > ### Author Response · Authors · 2021-11-19
> > **Response to Reviewer nwQT (part 2/2)**
> >
> > Our goal is to mimic the strategy developed in [1]  to build a two-staged algorithm to produce a low-rank approximation. We have the following key Lemma.
> > Using Lemma 3, we have
> >
> > $$\min_{rank(W)\leq k} ||\overline{Y}-\overline{X}W||_F^2        \quad s.t.\quad diag(W)=0$$
> >
> > $$ = \min_{rank(W) \leq k}  ||\overline{Y}-\overline{X} W^* ||_F^2 + ||\overline{X}W^* - \overline{X}W||_F^2       \quad s.t.  \quad diag(W^*)=0 \quad diag(W)=0 $$
> >
> > $$\ge \min_{rank(W) \leq k}  ||\overline{Y}-\overline{X} W^*||_F^2 + ||\overline{X}W^*-\overline{X}W||_F^2,  \quad s.t.  \quad diag(W^*)=0 $$
> >
> >
> > The last inequality holds because one constraint $diag(W) = 0$ is removed so the objective will only improve.
> > The closed-from solution (LR-EDLAE-1) is given by:
> > \begin{equation*}
> > \boxed{
> > \widehat{W}= W^*(Q_k Q_k^T)
> > }
> > \end{equation*} where $Q_k$ comes from:     $\overline{X}W^*\stackrel{\text{SVD}}{=}P \Sigma Q^T$.
> >
> >
> > **Intuition on the efficacy of the proposed approximation.**
> > We believe that because the full-rank $W^*$ performs well for recommendation accuracy and better than the low-rank solutions [1][2],  when $W$'s estimation is closer to $W^*$ with respect to the final recovery of $Y$, $XW \approx XW^*$, it has a comparable performance against $XW^*$.
> > In other words, it may not be essential for the diagonal of low-rank approximate matrix $\widehat{W}$ to be zero.
> >
> > Furthermore, if we consider the inequality
> > $$ \min_{rank(W) \le k}  ||\overline{X}W^* - \overline{X}W||_F^2$$
> >
> >  $$\le\min_{rank(W) \le k}  ||\overline{X}||_F^2||W^*-W||_F^2 $$
> >
> >
> > or a tighter one
> >
> > $$\min_{rank(W) \leq k}{||\overline{X}W^*-\overline{X}W||_F^2} $$
> >
> > $$ \leq\min_{rank(W) \leq k}{||\overline{X}||_2^2||W^*-W||_F^2}$$
> >
> > Both suggest if we simply perform SVD on $W^*$, we can obtain an upper bound error between $\bar{X}W^*$ and $\bar{X}W$.
> >
> > Simply, the other closed-form solution $\widehat{W}$ (LR-EDLAE-2) is: $W^*\stackrel{\text{SVD}}{=}U\Sigma V^T$, $\widehat{W} =U_k\Sigma_k V^T_k$
> >
> > **Q5. In Table 1, it would be clearer for the reader to identify the best methods for instance, 1) by putting in bold the best obtained results and 2) by underlying the 2nd best result. Table 1 refers to Table 3 for more details: I think this is a mistake because it does not let the paper "self-contained".**
> >
> > **A5.** We highlight the results in table 1 (now table 2 in revised version) based on the reviewer’s suggestions. Again we adjust the flow of the paper, put ahead the summary table (table 3 in originally submitted version now table 1) from appendix to introduction to make the paper "self-contained".
> >
> > **Q6. Am I missing something or Mult-VAE/DAE are used as baselines during experiments but are not discussed before (contrarily to the other approaches)? Why?**
> >
> > **A6.** In Section 2.1 (A4) together with Appendix D.3, we carefully analyze their linear counterpart, VLAE (Variational Linear Autoencoder). In fact, a starting point of the paper is trying to understand how Mult-VAE/DAE can be categorized based on the regularization (of course, after the linear simplification.)
> >
> > **References:**
> >
> > [1] Maurizio Ferrari Dacrema and P. Cremonesi and D. Jannach. "Are We Really Making Much Progress? A Worrying Analysis of Recent Neural Recommendation Approaches". RecSys'19
> >
> > [2] Bao, Xuchan, et al. "Regularized linear autoencoders recover the principal components, eventually." NeurIPS. 2020.
> >
> > [3] Ruoming Jin, et al. "Towards a better understandingof linear recommendation models." KDD 2021.
> >
> > [4] Steck, Harald. "Autoencoders that don't overfit towards the Identity." NeurIPS. 2020.

---

### Author Response · Authors · 2021-11-19
**General Response**

 We thank all the reviewers for their constructive feedback and comments, which help us improve our paper. Besides the individual response, we have significantly revised our paper by taking into account all your suggestions and questions. The main changes are summarized below:


   1.  **(Section 1: Introduction):** besides the first two paragraphs, we have completely rewritten the introduction to highlight the research problems,  our contributions, as well as the paper organization.
2.  **(Section 2:):** We have moved the Table 1 (originally in appendix) to Section 2, which summarizes
    different regularization methods and their solutions. Several parts are revised to highlight the regularizer dichotomy: the nuclear-norm and Frobenius-norm approaches.
3.  **(Section 3 \& 4):** minor revision to simplify the flow.
4.  **(Appendix B):** We have added a completely new section in Appendix to discuss the approximate closed-form solutions for low-rank EDLAE (in company with Section 4). We provide detailed mathematical derivation and rationals on deriving the closed-form solutions.
5.  **(Appendix C):** We have added a completely new section in Appendix to review the computational complexity of all closed-form solutions along with the ADMM solution on EDLAE.
6.  **(Notations and Typos):** We have also unified all the notations, terminologies as well as fixed all the typos that the reviewers have pointed out.

---

> ### Author Response · Authors · 2021-11-29
> **Thank you!!**
>
> Dear reviewers, we really appreciate all your questions,  critics and insights into our paper and the problems we are working on. They have helped reshape and improve  the paper significantly. If you have any other questions, concerns,  and comments, please let us know. We would like to provide our responses and address them in the future revision. Thank You!!

---

### Decision · Program_Chairs · 2022-01-20

**Decision:**

Reject

**Comment:**

The paper presents a new perspective on recommendation systems, categorizing them as linear predictors where the main difference between the various methods is the regularizer. The authors then propose an objective function that aims at optimizing the Frobenius norm while maintaining a low-rank solution, and present algorithm that have closed-form solutions based on the SVD.

The reviewers noted the novelty of the framework, but the overall assessment after the discussion was that the theoretical contribution was limited. The algorithm proposed by the authors does not provide any improvement on standard criteria (performance, computational complexity), which makes the algorithmic/experimental contribution limited as well.